# HumanEval-V: Evaluating Visual Understanding and Reasoning Abilities of Large Multimodal Models Through Coding Tasks

## Abstract

Coding tasks have been valuable for evaluating Large Language Models (LLMs), as they demand the comprehension of high-level instructions, complex reasoning, and the implementation of functional programs – core capabilities for advancing Artificial General Intelligence. Despite the progress in Large Multimodal Models (LMMs), which extend LLMs with visual perception and understanding capabilities, there remains a notable lack of coding benchmarks that rigorously assess these models, particularly in tasks that emphasize visual reasoning. To address this gap, we introduce `HumanEval-V`, a novel and lightweight benchmark specifically designed to evaluate LMMs' visual understanding and reasoning capabilities through code generation tasks. `HumanEval-V` includes 108 carefully crafted, entry-level Python coding tasks derived from platforms like CodeForces and Stack Overflow. Each task is adapted by modifying the context and algorithmic patterns of the original problems, with visual elements redrawn to ensure distinction from the source, preventing potential data leakage. LMMs are required to complete the code solution based on the provided visual context and a predefined Python function signature outlining the task requirements. Every task is equipped with meticulously handcrafted test cases to ensure a thorough and reliable evaluation of the model-generated code solutions. We evaluate 19 state-of-the-art LMMs using `HumanEval-V`, uncovering significant challenges. Proprietary models like GPT-4o achieve only 13% pass@1 and 36.4% pass@10, while open-weight models with 70B parameters score below 4% pass@1. Ablation studies further demonstrate the limitations of current LMMs in vision reasoning and coding abilities. These results highlight key areas for future research to enhance LMMs' capabilities.

## 1 Introduction

Coding ability is essential for both the development and evaluation of Large Language Models (LLMs) (Sun et al., 2024). By enabling LLMs to solve complex tasks in a divide-and-conquer manner, coding facilitates more autonomous and efficient interactions with the world (Patil et al., 2023; Liu et al., 2023b; Schick et al., 2024). As a result, coding tasks serve as a valuable testbed for advancing research in Artificial General Intelligence (Bubeck et al., 2023). Recently, Large Multimodal Models (LMMs) composed of billions of parameters have emerged, with notable examples such as GPT-4o (OpenAI, 2024) and Claude 3.5 Sonnet (Anthropic, 2024), demonstrating remarkable capabilities in understanding and reasoning within visual contexts.

While several recent multimodal benchmarks offer evaluations across a wide range of vision-related tasks (Goyal et al., 2017; Singh et al., 2019; Lu et al., 2022; Liu et al., 2023c; Yue et al., 2024), there remains a significant gap in benchmarks specifically designed for coding scenarios. These benchmarks typically involve multiple-choice or open-ended questions based on commonsense reasoning, neglecting more complex reasoning scenarios like coding. Notably, coding is a valuable form to assess complex reasoning abilities and has been exploited in various reasoning tasks such as mathematical, symbolic, and algorithmic reasoning (Madaan et al., 2022; Gao et al., 2023). It demands the ability to understand high-level instructions, apply complex logic, and implement functional programs. Moreover, coding enables a more robust evaluation of reasoning through program execution.

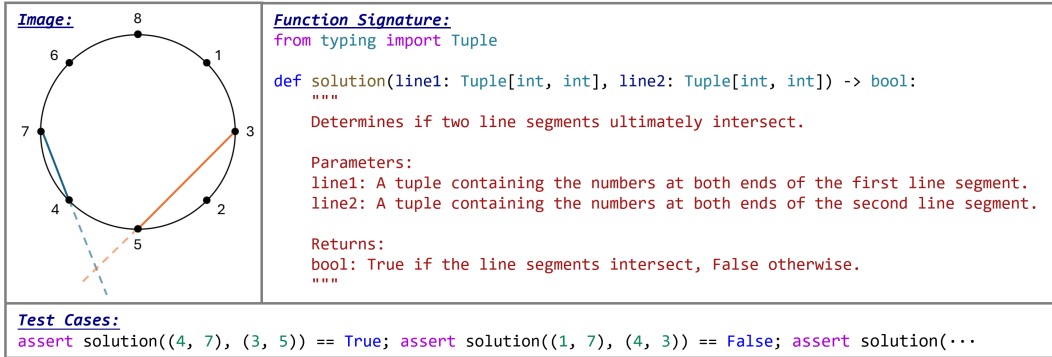

Figure 1: An example coding task in `HumanEval-V` that all LMMs evaluated in this work cannot solve, including GPT-4o and Claude 3.5 Sonnet. Related error analysis can be found in Appendix A.

To address this gap, we introduce `HumanEval-V`, a novel and lightweight benchmark tailored to evaluate LMMs in coding scenarios. `HumanEval-V` consists of 108 manually crafted code generation tasks sourced from platforms such as CodeForces and Stack Overflow. Each task is adapted from the source by carefully modifying the original problem's context and algorithmic patterns as well as redrawing the visual elements. As an example task shown in Figure 1, each task involves completing a Python function based on a single image, a function signature, and problem descriptions provided in the comment block. These tasks require reasoning over both visual and textual contexts to complete a function, with the correctness of the predicted solution assessed using a reliable set of human-annotated test cases.

`HumanEval-V` is novel in that it is the first benchmark where *visual information plays an essential role in solving coding tasks*. For instance, the diagram in Figure 1 not only indicates the available position options for the function inputs, but also offers important clues for determining whether two lines intersect, significantly complementing the function signature and problem descriptions. To solve these tasks, models have to accurately understand the nuances of the image, such as the position of two lines on the circle and tick labels. Moreover, they need the ability to perform cross-modal reasoning, integrating visual elements with the structured function signature and textual problem descriptions cohesively. In contrast to other benchmarks (Li et al., 2024b), which suggest that visual information has limited impact on coding performance, `HumanEval-V` ensures that all coding tasks are unsolvable without the visual context. Textual descriptions in the coding tasks are minimized to prevent models from relying solely on textual information to infer solutions.

Another appealing characteristic of `HumanEval-V` is *light-weight and easy to test*. It mirrors the difficulty of well-established code generation benchmarks like HumanEval (Chen et al., 2021) and MBPP (Austin et al., 2021) that target entry-level programmers. The simplicity of evaluation has been one of the key reasons for the wide adoption of these benchmarks. In `HumanEval-V`, each task is formulated in a Python code completion setting like HumanEval and annotated with a comprehensive suite of test cases in a format of assertion statements, making it easy to execute and efficient to measure the correctness of the completion. Additionally, the tasks are restricted to using only common Python libraries, promoting the accessibility without requiring domain-specific knowledge and avoiding compatibility issues with different library versions. We perform cross-validation among several annotators to ensure the data integrity.

Through extensive experiments with 19 state-of-the-art LMMs, we have the following key findings: (1) Even leading proprietary models like GPT-4o achieve only 13% pass@1 on `HumanEval-V`, while open-weight models perform much worse, with none of them exceeding 4% pass@1. `HumanEval-V` reveals limitations of current LMMs. (2) Proprietary models significantly outperform open-weight LMMs, highlighting the challenges in developing more advanced open-weight models. (3) Current LMMs remain limited in their visual reasoning abilities, as evidenced by the significant performance gains when provided with human-annotated textual descriptions of the images. (4) Open-weight LMMs suffer from deteriorated coding performance after integrating the vision encoder. These findings emphasize the need for future research to enhance LLMs' visual reasoning and coding abilities.

## 2 BENCHMARK CONSTRUCTION

As shown in Figure 1, each coding task in `HumanEval-V` consists of three main components. The first component is a single image input, denoted as $I$, which provides the essential visual context necessary to solve the coding problem. The second component is a Python function signature, denoted $\sigma$, which specifies the function name, input parameters, and return type, accompanied by a brief problem description in the comment block. Both the image $I$ and the function signature $\sigma$ are formatted into a predefined prompt template, which is then provided to the LMM. The model's output, denoted as $O$, represents the complete Python function generated by the LMM based on $\sigma$ and $I$. The third component is a set of test cases $T = \{t_1, t_2, \ldots, t_n\}$, which are used to validate the functional correctness of $O$ through execution. A solution is considered correct if $O$ passes all test cases, meaning it produces the expected outputs for each $t_i \in T$.

Before constructing `HumanEval-V`, we establish rigorous standards to ensure the quality of the coding task annotations: (1) the visual context provided must be essential for solving the task, with all relevant information contained within a single image; (2) the coding task should be largely self-explanatory through its visual context, requiring minimal textual descriptions; and (3) the coding task should target entry-level programmers and be solvable using only common Python libraries.

The construction of `HumanEval-V` follows a collect-adapt-mutate pipeline. First, we collect coding problems with visual contexts from platforms such as CodeForces and Stack Overflow, identifying those suitable for adaptation based on the aforementioned standards. (Section 2.1). Next, we modify the selected problems by adapting their task descriptions and redrawing the visual elements to ensure they meet our quality requirements. During this stage, we annotate each task with a function signature ($\sigma$), a set of test cases ($T$), and a ground truth solution. To further expand the benchmark, some tasks undergo mutations, generating similar yet distinct versions by introducing changes to the coding task's visual patterns while preserving the core context. This iterative process results in a final set of 108 code generation tasks (Section 2.2). After constructing the benchmark, we perform rigorous validation to ensure that each coding task aligns with the standards: testing visual reasoning, preventing data leakage, and maintaining an appropriate entry-level complexity. Finally, we provide detailed benchmark statistics for reference (Section 2.3).

### 2.1 DATA COLLECTION AND SCREENING

The coding tasks in `HumanEval-V` are sourced from prominent Q&A and coding challenge platforms such as Stack Overflow and CodeForces. These platforms offer a diverse range of coding problems and are also commonly used in the development of well-established benchmarks for code generation (Yin et al., 2018; Lai et al., 2023; Wang et al., 2023; Li et al., 2023b; Jain et al., 2024; Wu et al., 2024b). From these sources, we collect a large set of coding problems that incorporate visual elements in their problem descriptions.

However, the collected problems are unsuitable for direct inclusion in `HumanEval-V`. In most cases, the visual context is non-essential for solving the task, with the problem primarily solvable through rich textual descriptions alone. This makes it challenging to adapt such problems into our benchmark, which emphasizes visual reasoning abilities. Therefore, we focus on identifying tasks that already feature high-quality visual elements and present a moderate level of difficulty. After a thorough screening process, we retain 40 candidate coding tasks out of the thousands reviewed for further adaptation. A detailed discussion of the challenges encountered during data collection and screening, along with demonstrating examples, is provided in Appendix C.1.

### 2.2 CODING TASK ANNOTATION

The annotation process begins by adapting the screened coding problems. For each of the 40 selected coding tasks, we first identify and summarize the essential context and algorithmic patterns required to solve the problem. We then create a new coding problem by modifying the context and patterns of the original problem and redrawing the corresponding images. This is to prevent data leakage and ensure consistency with the standards of `HumanEval-V`. Detailed examples of the problem adaptation can be found in Appendix C.2.

During adaptation, we ensure that all critical visual information for each coding task is encapsulated within a single image. The coding tasks in `HumanEval-V` span a variety of visual elements, including trees, graphs, matrices, maps, grids, flowcharts, and other abstract representations. This diversity allows for comprehensive testing of the model's visual reasoning abilities. Next, we define a Python function signature for each task, beginning with the input and output specifications. To simplify the Input/Output (I/O) formats, we prioritize basic data structures such as numbers, strings, lists, and dictionaries. After finalizing the image and I/O definitions, we craft a concise problem description that directs models to rely primarily on the visual information to complete the Python function. Once the task definition is completed, we manually construct test cases and implement a ground truth solution for each coding problem to ensure its validity. To further verify the comprehensiveness of the test cases, we perform statement and branch coverage analysis on the ground truth solution, ensuring that all logical branches and execution paths are thoroughly tested.

Following the initial annotation of the 40 coding tasks, we conduct an additional round of mutation-based extensions. This process expands the number of coding tasks based on the initial annotations, by creating similar yet distinct coding tasks. The mutated tasks retain most of the original visual elements but incorporate different algorithms to solve. For example, we can change the rule of the coding task in Figure 1 by just considering the situation where the line segments intersect within the circle, regardless of outside the circle. It is important to note that not all of the 40 tasks are suitable for mutation. For each suitable task, we create one or two mutations, resulting in a total of 108 coding tasks in `HumanEval-V`. Examples of the mutation process are provided in Appendix C.3

## 2.3 QUALITY ASSURANCE AND DATASET STATISTICS

We implement a rigorous quality assurance process to ensure the quality of `HumanEval-V`. The annotation team consists of three programmers, each with over four years of Python programming experience. During each of the data collection, adaptation, and mutation stages, annotators independently perform annotations based on pre-defined guidelines. After that, all annotators conduct a cross-validation process to review and resolve any identified issues. A coding task is only finalized when consensus is reached among all annotators. Additionally, one annotator maintains consistent formatting and style across all visual representations and coding tasks. Each annotator dedicates over 200 hours to the overall benchmark construction process. To validate the reliance on visual context, we ensure that GPT-4o cannot solve any of the coding tasks without access to the images, confirming the essential role of visual information. Finally, to facilitate continuous improvement, we will publish an online data viewer for `HumanEval-V` after the review period, where the community can review the dataset and report issues.

| Attributes | Med | Avg | Min | Max |
|---|---|---|---|---|
| Image Width (px) | 1024 | 998.2 | 596 | 1024 |
| Image Height (px) | 709 | 729.0 | 216 | 1024 |
| Textual Token Count | 106 | 111.3 | 59 | 230 |
| GT Code Statements | 14 | 16.3 | 3 | 44 |
| Test Cases Count | 10 | 9.8 | 4 | 16 |

Table 1: The descriptive statistics for the key attributes of `HumanEval-V`, showcasing the Median, Average, Minimum, and Maximum values.

To provide a clearer understanding of our benchmark, Table 1 presents key statistics for several dataset attributes. Each coding task includes a single image input, with the image dimensions constrained to a maximum of 1024 pixels in height or width, to prevent overly long or complex visual contexts. The average image width and height are 998.2 and 729 pixels, respectively. We also analyze the length of function signatures using the OpenAI *tiktoken*[1] tokenizer. The longest function signature consists of 230 tokens, while the average token count is 111.3, demonstrating high succinctness. We also quantify the complexity of the ground truth (GT) code solutions annotated by human experts. On average, GT solutions contain 16.3 code statements, encompassing import statements, function definitions, and the function body, reflecting the relative simplicity of the tasks. Finally, we provide statistics on the number of test cases used for evaluation, with an average of 9.8 test cases per task. We ensure the test cases achieve full statement and branch coverage on the GT solutions, guaranteeing rigorous testing of the generated code. We also include a detailed list of the I/O types and module dependencies in Appendix C.4.

---

[1] `https://github.com/openai/tiktoken`

## 3 EXPERIMENTAL SETUP

**Models:** We conduct a comprehensive evaluation of 19 state-of-the-art LMMs to assess the current progress in visual reasoning and coding capabilities. Our selection includes a representative set of the most advanced proprietary and open-weight models. Specifically, we evaluate five of the latest proprietary models: GPT-4o (0513), GPT-4o-mini (0718) (OpenAI, 2024), Gemini 1.5 Pro (001), Gemini 1.5 Flash (001) (Google, 2024), and Claude 3.5 Sonnet (0620) (Anthropic, 2024). In addition, we test 14 top-performing open-weight models, selected based on their rankings on the OpenVLM Leaderboard (Duan et al., 2024). These models span various parameter sizes to explore the impact of scale on performance in the `HumanEval-V` benchmark. The open-weight models include Phi-3-Vision (4.2B) (Microsoft, 2024a), Phi-3.5-Vision (4.2B) (Microsoft, 2024b), LLaVA-OneVision (8.0B, 73.2B) (Li et al., 2024a), MiniCPM-V 2.5 (8.5B) and 2.6 (8.1B) (Yao et al., 2024b), InternVL-Chat-V1.5 (26.0B) (Chen et al., 2023), InternVL-2 (4.2B, 8.1B, 25.5B, 40.1B) (OpenGVLab, 2024), and Qwen2-VL (8.3B, 73.4B) (Wang et al., 2024). We deliberately include different versions within the same model series, such as Phi-3-Vision and Phi-3.5-Vision, MiniCPM-V 2.5 and 2.6, as well as InternVL-Chat-V1.5 and InternVL-2, to investigate whether iterative improvements in model development result in enhanced performance on `HumanEval-V`. More details of the models can be found in Appendix D.

```
**Instructions:**
You are an exceptionally intelligent
coding assistant that consistently
delivers accurate and reliable responses
to user instructions. Please complete
the function based on the provided image
and code context. Return the complete
solution, including the function
signature, in a single response,
formatted within a Python code block.

**Code Context:**
```python
{code_context}
```
```

Figure 2: The prompting template used for LMMs to generate code solutions. The {code_context} placeholder is for the corresponding function signature.

**Prompting, Hyper-parameters, and Post-processing:** All the LMMs evaluated in our experiments have been trained on instruction-following or conversational data. To align with this, we employ a conversational prompt template, formatted in Markdown, as illustrated in Figure 2, to prompt the LMMs to generate code solutions for the tasks in `HumanEval-V`. For hyper-parameters, we follow the established practices in code generation benchmarking (Chen et al., 2021; Austin et al., 2021; Chen et al., 2022), using two distinct settings. First, we employ greedy search to generate a single code solution from each LMM, allowing us to assess the models' performance in a deterministic setting. Additionally, we sample 20 code solutions using a Top-$p$ sampling method with $p = 0.95$ and a relatively high temperature of 0.8. This setting is designed to explore the likelihood of the models generating correct solutions when given more opportunities. Given the moderate complexity of the benchmark, we set the maximum number of new tokens for code generation to 1024. Early stopping is triggered by "\n```\n", since the LMMs are instructed to enclose the generated code within a Markdown code block. We also develop a post-processing pipeline to extract valid code solutions from the model outputs. This pipeline identifies and extracts the content within the Markdown code block and uses an abstract syntax tree parser to detect any generated import statements, along with class and function definitions. These components are then concatenated to form the final predicted solution for test-execution-based evaluation.

**Evaluation Metrics** Following established practices in code generation (Chen et al., 2021; Austin et al., 2021; Chen et al., 2022), we use the pass@$k$ metric to evaluate the functional correctness of the generated code solutions. For each coding task, $n$ code samples are generated, and $k$ solutions are randomly selected from these samples to be tested against ground truth test cases. A task is considered solved if at least one of the $k$ selected solutions passes all test cases. The pass@$k$ score is then calculated as the percentage of successfully solved tasks. In our main experiments, we report pass rate results for $k = 1, 10$. For greedy search, we set $n = 1$ to compute pass@1. For sampling-based evaluation, we set $n = 20$ to calculate pass@10.

We incorporate a second evaluation metric: *Execution Success Rate*. This metric measures the syntactic correctness of the generated code, independent of its functional accuracy. A solution is considered executable if it can be compiled and run without triggering syntax errors, null pointer exceptions, or other runtime failures, regardless of passing the test cases. The execution success rate is calculated as the proportion of executable code samples out of all generated samples.

# 4 EXPERIMENTAL RESULTS

## 4.1 MAIN EXPERIMENTS

| LMMs | Params | Exec. Rate | pass@k | |
| --- | --- | --- | --- | --- |
| | | | k=1 | k=10 |
| Proprietary | | | | |
| GPT-4o | | 87.9 | 13.0 | **36.4** |
| GPT-4o-mini | | 90.4 | 6.5 | 15.4 |
| Claude 3.5 Sonnet | | 91.8 | **18.5** | 25.9 |
| Gemini 1.5 Pro | | 92.9 | 10.2 | 22.2 |
| Gemini 1.5 Flash | | 92.6 | 8.3 | 13.2 |
| Open-Weight | | | | |
| InternVL-2 | 76.3B | 72.8 | **3.7** | 12.8 |
| | 40.1B | 66.2 | 0.0 | 1.6 |
| | 25.5B | 57.8 | 0.0 | 3.2 |
| | 8.1B | 64.6 | 0.9 | 2.6 |
| | 4.2B | 76.5 | 0.0 | 2.3 |
| Qwen2-VL | 73.4B | 86.3 | **3.7** | **16.0** |
| | 8.3B | 58.1 | 0.0 | 1.6 |
| LLaVA-OneVision | 73.2B | 84.7 | 1.9 | 12.4 |
| | 8.0B | 69.6 | 0.9 | 1.9 |
| InternVL-Chat-V1.5 | 25.5B | 62.0 | 0.0 | 2.1 |
| MiniCPM-V 2.6 | 8.1B | 67.2 | 0.9 | 2.2 |
| MiniCPM-V 2.5 | 8.5B | 75.7 | 0.0 | 2.3 |
| Phi-3.5-Vision | 4.2B | 75.0 | 0.9 | 1.6 |
| Phi-3-Vision | 4.2B | 76.1 | 0.0 | 2.6 |

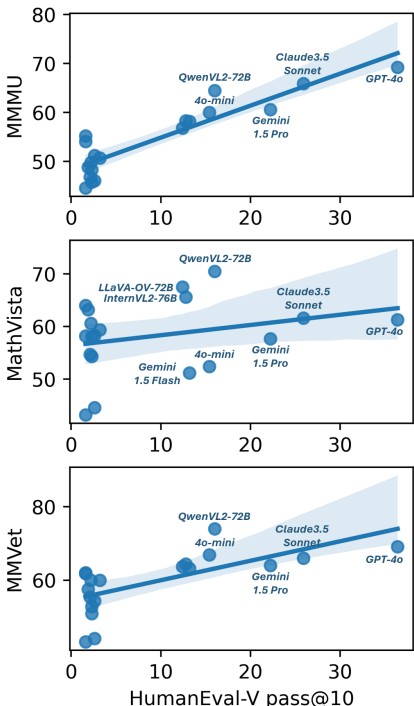

Table 2: Performance of 19 LMMs on `HumanEval-V`. Scores are shown as percentages, with the highest values highlighted in **bold**. We also include model size (Params) and code execution success rate (Exec. Rate).

Figure 3: Correlation analysis between `HumanEval-V` pass@10 results and three popular multimodal benchmarks spanning multidisciplinary abilities.

We evaluate 19 state-of-the-art LMMs on `HumanEval-V`, with results presented in Table 2. Based on the results, we have the following key findings:

**Current LMMs' performance is underwhelming on our benchmark:** While proprietary models like GPT-4o and Claude 3.5 Sonnet show the best results, even their highest pass@1 scores (13% and 18.5% respectively) fall short of expectations. Moreover, there remains a substantial performance gap between proprietary and open-weight models. Open-weight models spanning 4B to 76B parameters exhibit particularly weak performance, with none exceeding a 4% pass@1. This is surprising given that the coding tasks in our benchmark are designed for entry-level programmers with simplified problem context. None of the open-weight models with fewer than 70B parameters achieve more than 4% pass@10. Even the best-performing model, GPT-4o, achieves only 36.4% pass@10, showing there is much room for improvement. In terms of execution success rate, we observe a rough correlation with the pass rate. Most LMMs exhibit a high execution success rate, while smaller-scale open-weight models show lower success rates. Most failed cases are due to common syntax errors, such as unclosed parentheses, generating code repeatedly without termination, or encountering list index out-of-range issues. To further investigate, we perform another experiment increasing the number of samples to evaluate model performance, as detailed in Appendix B.1.

**Overfitting leads to hallucination errors in LMMs' generated solutions:** Upon examining many incorrect solutions produced by the LMMs, we identify a recurring issue: the models tend to generate solutions based on the context of the original problems rather than the new versions of coding tasks in our benchmark. For instance, both GPT-4o and Claude 3.5 Sonnet fail to produce correct solutions for the coding task shown in Figure 1, as they mistakenly assume that the numbers in the image are arranged in a clockwise order. Furthermore, their solutions rely on the assumption that the two line segments can only intersect within the circle, which reflects the context of the original

| Models | Params | Image Only | | Desc. Only | | Image & Desc. | |
|---|---|---|---|---|---|---|---|
| | | pass@1 | pass@10 | pass@1 | pass@10 | pass@1 | pass@10 |
| *Large Multimodal Models* | | | | | | | |
| GPT-4o | | 13.0 | 36.4 | 45.4↑32.4 | 67.9↑31.6 | 44.4↑31.5 | 71.0↑34.6 |
| GPT-4o-mini | | 6.5 | 15.4 | 33.3↑26.9 | 46.1↑30.7 | 35.2↑28.7 | 50.6↑35.2 |
| InternVL-2 | 76.3B | 3.7 | 12.8 | 12.0↑8.3 | 41.1↑28.3 | 23.2↑19.5 | 47.9↑35.1 |
| | 25.5B | 0.0 | 3.2 | 2.8↑2.8 | 15.7↑12.5 | 4.6↑4.6 | 15.2↑12.0 |
| | 8.1B | 0.9 | 2.6 | 3.7↑2.8 | 10.3↑7.8 | 5.6↑4.6 | 12.3↑9.7 |
| | 4.2B | 0.0 | 2.3 | 5.6↑5.6 | 16.2↑13.9 | 2.8↑2.8 | 13.0↑10.7 |
| Qwen2-VL | 73.4B | 3.7 | 16.0 | 20.4↑16.7 | 38.9↑22.9 | 23.2↑19.5 | 48.2↑32.2 |
| | 8.3B | 0.0 | 1.6 | 5.6↑5.6 | 13.5↑11.9 | 3.7↑3.7 | 16.9↑15.2 |
| MiniCPM-V 2.6 | 8.1B | 0.9 | 2.2 | 3.7↑2.8 | 7.1↑4.8 | 2.8↑1.9 | 6.9↑4.6 |
| MiniCPM-V 2.5 | 8.5B | 0.0 | 2.3 | 0.9↑0.9 | 14.6↑12.2 | 2.8↑2.8 | 14.2↑11.9 |
| Phi-3.5-Vision | 4.2B | 0.9 | 1.6 | 0.0↓0.9 | 9.8↑8.2 | 2.8↑1.9 | 10.0↑8.3 |
| Phi-3-Vision | 4.2B | 0.0 | 2.6 | 3.7↑3.7 | 10.0↑7.5 | 2.8↑2.8 | 6.8↑4.3 |
| *Large Code Language Models* | | | | | | | |
| CodeStral | 22.2B | | | 18.5 | 36.8 | | |
| DSCoder-V2-Lite | 15.7B | | | 13.0 | 37.4 | | |
| Yi-Coder-Chat | 8.8B | | | 25.0 | 40.2 | | |
| DSCoder-V1.5 | 6.9B | | | 13.0 | 21.5 | | |

Table 3: The performance of LMMs and Code LLMs on `HumanEval-V` using different input settings. "*Image Only*" refers to the setting used in the main experiments. "*Desc. Only*" evaluates models using annotated descriptions of images instead of the images themselves. "*Image & Desc.*" provides both inputs to the models. Scores are presented as percentages. The ↑ and ↓ indicate performance improvement and degradation over the "*Image Only*" setting.

problem on the CodeForces platform, rather than our adapted version. We attribute these hallucination errors to that LMMs overfit on the previously seen data. This observation underscores the necessity of our adaptation process, which aims to minimize data leakage and prevent models from relying on memorized patterns.

**Larger parameter size does not guarantee better performance in open-weight models:** While open-weight LMMs with over 70B parameters show superior results, smaller models (ranging from 4B to 40B parameters) exhibit highly variable performance. For example, Phi-3-Vision (4.2B) and InternVL-2 (4.2B) achieve pass@10 scores of 2.6% and 2.3%, outperforming larger models like QwenVL2 (8.3B) and InternVL-2 (40.1B). Notably, iterations of the Phi-Vision (3→3.5) and MiniCPM-V(2.5→2.6) series do not lead to consistent performance improvements. This inconsistency may be attributed to several factors. One possibility is the varying quality and scale of the training data used for each model, which can impact their generalization ability.

**Our benchmark reveals unique weaknesses in LMMs:** Open-weight LMMs, such as Qwen2-VL (Wang et al., 2024) and InternVL-2 (OpenGVLab, 2024), have demonstrated comparable or even superior performance to proprietary LMMs on popular multimodal benchmarks like MMMU (Yue et al., 2024), MathVista (Lu et al., 2023), and MMVet (Yu et al., 2023). However, these models perform significantly worse on `HumanEval-V`, suggesting that our benchmark exposes previously undetected limitations in current LMMs. The three mentioned benchmarks evaluate a broad range of multidisciplinary abilities, focusing on visual understanding, reasoning, and general knowledge through formats such as question-answering or multiple-choice questions, using accuracy as the primary evaluation metric. By contrast, `HumanEval-V` adopts a unique evaluation approach based on coding tasks, where visual contexts are tightly integrated with algorithmic patterns, presenting a distinct challenge that differs from existing benchmarks. To further investigate this discrepancy, we perform a correlation analysis between `HumanEval-V` and the three mentioned benchmarks.

We collect the performance results of the 19 evaluated LMMs from the OpenVLM Leaderboard (Duan et al., 2024) as well as from corresponding papers and reports, and compare them to pass@10 scores on `HumanEval-V` in a regression plot, shown in Figure 3. For proprietary models, we observe a rough positive correlation between `HumanEval-V` and the other benchmarks.

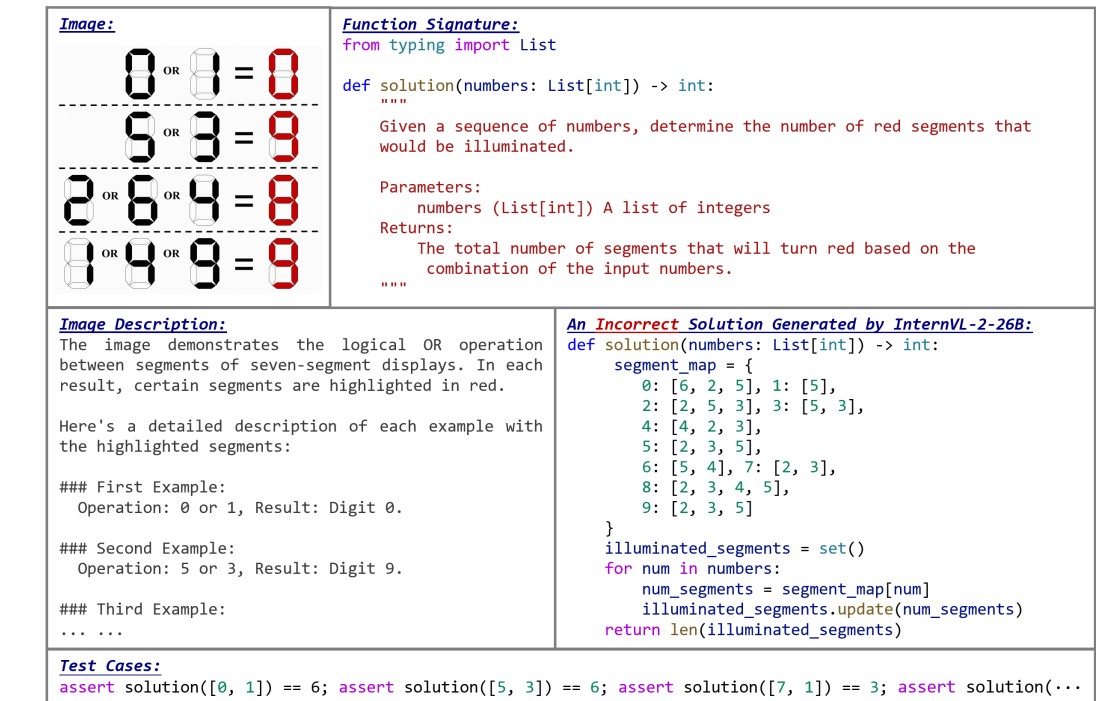

Figure 4: A coding task that InternVL-2-26B fails to solve with grounded image description.

While many scatter points for our benchmark are concentrated around zero, even though they show competitive results on the other three benchmarks, highlighting the distinct challenge posed by our benchmark. A comprehensive analysis of correlations between `HumanEval-V` and 5 other benchmarks can be found in Appendix B.2.

## 4.2 ANALYSING EXPERIMENTS

To investigate the reasons behind the suboptimal performance of current LMMs on `HumanEval-V`, we perform analyzing experiments by answering two key research questions.

**Q1. Are LMMs Limited by Their Vision Capabilities?**

We conduct an ablation study to evaluate whether the limitations in visual understanding contribute to the underperformance of LMMs. In this study, we manually annotate detailed descriptions for each image in the coding tasks, ensuring that these descriptions are descriptive rather than instructive, without revealing any specific algorithms. We design a new prompt template incorporating the image description to provide LMMs with better-grounded visual context, thereby reducing issues such as ambiguity and hallucination. Details of the new prompt template and examples of annotations are provided in Appendix B.3. To further assess the quality of our annotations, we also test a setting where LMMs receive only the image descriptions, without access to the images themselves. Additionally, we evaluate several top-performing Code LLMs using image descriptions to explore their potential in `HumanEval-V`. We present the results in Table 3. Below are the key findings:

(1) The inclusion of image descriptions leads to notable performance gains across all LMMs, with higher-capability models showing the most significant improvements. For example, GPT-4o exhibits a 31.5% absolute increase in pass@1. Similarly, large open-weight LMMs demonstrate substantial improvement, indicating that current models still require enhanced visual understanding capabilities. However, the limited improvement observed in smaller open-weight models suggests that merely perceiving visual elements is insufficient for solving tasks that require deeper reasoning. We illustrate this limitation with an example from InternVL-2 (25.5B) shown in Figure 4. The task requires determining the number of illuminated red segments based on an "OR" operation depicted in the image. While the model's solution correctly implements the algorithm, it fails to identify

| LMMs | LLM Decoders | Params | | HumanEval$^+$ | | MBPP$^+$ | |
|---|---|---|---|---|---|---|---|
| | | LLM | LMM | LLM | LMM | LLM | LMM |
| InternVL-2 | Nous-Hermes-2-Yi | 34.4B | 40.1B | 66.5 | 38.4$_{\downarrow 28.1}$ | 57.9 | 47.1$_{\downarrow 10.8}$ |
| InternVL-2 | InternLM2-Chat | 19.9B | 25.5B | 65.2 | 54.9$_{\downarrow 10.3}$ | 55.4 | 51.9$_{\downarrow 3.5}$ |
| InternVL-2 | InternLM2.5-Chat | 7.7B | 8.1B | 63.4 | 50.0$_{\downarrow 13.4}$ | 53.9 | 52.4$_{\downarrow 1.5}$ |
| InternVL-2 | Phi-3-Mini-Instruct | 3.8B | 4.2B | 64.0 | 57.3$_{\downarrow 6.7}$ | 57.1 | 57.1$_{0.0}$ |
| Phi-3.5-Vision | Phi-3.5-Mini-Instruct | 3.8B | 4.2B | 65.9 | 51.8$_{\downarrow 14.1}$ | 52.6 | 50.4$_{\downarrow 2.2}$ |
| Qwen2-VL | Qwen2 | 7.6B | 8.3B | 58.5 | 65.2$_{\uparrow 6.7}$ | 53.1 | 43.6$_{\downarrow 9.5}$ |
| LLaVA-OneVision | Qwen2 | 7.6B | 8.0B | 58.5 | 59.1$_{\uparrow 0.6}$ | 53.1 | 51.6$_{\downarrow 1.5}$ |
| MiniCPM-V 2.6 | Qwen2 | 7.6B | 8.1B | 58.5 | 39.6$_{\downarrow 18.9}$ | 53.1 | 37.6$_{\downarrow 15.5}$ |
| MiniCPM-V 2.5 | Llama-3-Instruct | 8.0B | 8.5B | 55.5 | 46.3$_{\downarrow 9.2}$ | 51.9 | 47.1$_{\downarrow 4.8}$ |
| GPT-4o | | | | | 86.0 | | 68.7 |
| GPT-4o-mini | | | | | 84.8 | | 65.7 |

Table 4: The performance comparison of open-weight LMMs and their corresponding LLM decoders on HumanEval$^+$ and MBPP$^+$ benchmarks. Scores are shown as percentages, with ↑ and ↓ indicating performance improvement and degradation of LMMs compared to their LLM decoders.

the segment mappings for each number, as this information is not explicitly provided in the image description. This example underscores the challenge of integrating visual and textual reasoning in coding tasks. (2) The "*Desc. Only*" setting performs comparably to the "*Image & Desc.*" setting, underscoring that the annotated image descriptions can effectively capture the key visual information to solving the task. (3) The Code LLMs with small-scale parameter sizes perform well on the tasks when provided with image descriptions alone (i.e., without access to the images). For instance, Yi-Coder-Chat (8.8B) achieves a 25% pass@1 and a 40.2% pass@10. This highlights the great potential for current open-weight LMMs to further develop their reasoning and coding abilities.

**Q2. Are LMMs Limited by Their Coding Abilities?**

Open-weight LMMs with parameter sizes ranging from 4B to 40B exhibit surprisingly low performance on `HumanEval-V`, even when utilizing grounded visual elements through image descriptions. This suggests that open-weight LMMs may suffer from degradation of relevant coding abilities. So we evaluate the models on a well-established code generation benchmark, EvalPlus Liu et al. (2023a), to investigate their coding abilities. This benchmark includes two sub-datasets refined from HumanEval (Chen et al., 2021) and MBPP (Austin et al., 2021), both consisting of Python function completion tasks with problem descriptions and test-execution-based evaluation. Different from `HumanEval-V`, these datasets depend exclusively on textual context.

Given that open-weight LMMs typically employ a vision-encoder and language-decoder architecture, we also evaluate their LLM decoders separately to determine whether their coding performance deteriorates after integrating the vision abilities. The results presented in Table 4 lead to the following findings: (1) Open-weight LMMs consistently experience performance degradation on coding benchmarks compared to their LLM decoders, despite having similar parameter sizes. Among these, InternVL-2 (40.1B) and MiniCPM-V 2.6 show the most degradation, while InternVL-2 (4.2B) and LLaVA-OneVision (8B) show the least. (2) Despite this degradation, open-weight LMMs still exhibit relatively strong coding capabilities. Although their performance on EvalPlus does not match GPT-4o, many of these models produce competitive results, indicating they retain a substantial degree of code generation ability. These results highlight the need for further improvement in the coding abilities of current open-weight LMMs.

## 5 RELATED WORK

While numerous benchmarks have been developed to evaluate various capabilities of LMMs, ranging from optical character recognition (OCR) to multidisciplinary knowledge reasoning, few specifically focus on the intersection of visual reasoning and code generation. This section reviews the current progress of LMM benchmarking and demonstrates how `HumanEval-V` fills this gap.

**OCR and Multidisciplinary Knowledge Abilities:** A variety of benchmarks have been developed to evaluate multidisciplinary capabilities of LMMs. There are popular benchmarks like

DocVQA (Mathew et al., 2021), ChartQA (Masry et al., 2022), TextVQA(Singh et al., 2019), OCR-Bench (Liu et al., 2023d), and OCRVQA (Mishra et al., 2019) assess models' ability to recognize and interpret text embedded in visual formats, including documents, charts, and images, often combining these with multiple-choice questions (MCQ) and visual question answering (VQA) tasks. Meanwhile, benchmarks such as MMMU (Yue et al., 2024), MME (Fu et al., 2023), MMBench (Liu et al., 2023c), MMVet (Yu et al., 2023), SEEDBench (Li et al., 2023a), MMT-Bench (Ying et al., 2024), and MMStar (Chen et al., 2024) test models on their general knowledge and reasoning abilities across diverse domains, such as scientific concepts, cultural knowledge, and logical reasoning. In contrast, `HumanEval-V` distinguishes itself by expanding the evaluation format beyond traditional MCQ and VQA. `HumanEval-V` requires models to interpret visual elements and apply that understanding to generate correct and executable code, which introduces a more complex challenge.

**Specialized Abilities:** There are also benchmarks focusing on specific capabilities of LMMs. MathVista (Lu et al., 2023) evaluates mathematical problem-solving skills. Safety-related benchmarks (Gu et al., 2024) assess models on their ability to recognize and mitigate potential risks or harmful content. ConvBench (Liu et al., 2024) evaluates conversational abilities, testing models on their proficiency in maintaining coherent and contextually relevant dialogues. Benchmarks for instruction-following ability (Qian et al., 2024) assess how well models can execute tasks based on given instructions. Long-context reasoning benchmarks (Ma et al., 2024) assess the ability of models to maintain coherence and logical reasoning over extended dialogues or documents. HallusionBench (Guan et al., 2024) focuses on hallucination detection abilities to differentiate between factual information and generated content. There are also benchmarks (Zhang et al., 2024) evaluating mobile app navigation, testing models on their ability to interpret and interact with user interfaces. In contrast, `HumanEval-V` mainly focuses on integrating visual reasoning and coding.

**Coding Abilities:** Despite the wide range of benchmarks available, the coding ability of LMMs remains under-explored. Coding capabilities are crucial for leveraging LMMs in autonomous and agentic applications (Xie et al., 2024). Current efforts focus primarily on derendering web pages (Si et al., 2024; Laurençon et al., 2024) and scientific figures (Shi et al., 2024; Wu et al., 2024a), where models translate visual representations into code. The other related area is Program-based VQA, where models are provided with a set of pre-defined modules (e.g., for OCR, object detection, and segmentation) and tasked with invoking them to answer visual questions like counting or identifying spatial relationships (Surís et al., 2023; Subramanian et al., 2023). These methods show how models can use existing tools to perform vision tasks, while they complicate evaluation due to reliance on multiple heavy modules. In contrast, `HumanEval-V` utilizes simple Python coding tasks to streamline evaluation and focuses on visual understanding in coding tasks. Another closely related work is MMCode (Li et al., 2024b), which evaluates the coding ability of LMMs on visually rich competition-level coding problems. utilizing existing coding challenges from competitive programming websites. However, MMCode overlooks two critical issues: the potential for data leakage when relying on scraped data, and the use of text-rich problem contexts, which makes visual information non-essential for solving tasks. By contrast, our approach addresses both concerns with rigorous data screening and annotation. We list a detailed discussion on MMCode in Appendix E.

## 6 CONCLUSION

We present a novel and lightweight benchmark `HumanEval-V` designed to evaluate the visual reasoning capabilities of LMMs through 108 high-quality, entry-level Python coding tasks that rely on visual context to solve. We ensure rigorous evaluation of generated code solutions using comprehensive test cases. Our benchmark effectively uncovers weaknesses in current LMMs that are overlooked by existing benchmarks. Through our analysis, we identify three critical limitations in the current generation of LMMs. First, their visual perception abilities remain inadequate. We observe significant performance gains when we provide textual descriptions of visual elements, indicating that models still struggle to understand visual context independently. Second, open-weight LMMs exhibit a consistent decline in their coding proficiency compared to their LLM decoders, suggesting that the current multimodal training strategy still needs improvement. Finally, hallucination due to overfitting is a major issue, causing models to incorrectly apply memorized patterns rather than adapt to the new visual context in the coding tasks. We hope these findings will inform and guide future research on enhancing the visual reasoning and coding capabilities of LMMs. We also provide a discussion on our work's limitations in Appendix F.

REPRODUCIBILITY STATEMENT

We provide all code and data in a publicly available anonymous repository (`https://anonymous.4open.science/r/HumanEval-V-Anonymous/`) for reference. The repository includes the full dataset for our benchmark, accompanied by detailed usage instructions. Our source code contains all the necessary components for running model inference to generate code solutions, as well as evaluation scripts for obtaining and analyzing the results. Additionally, we provide setup guides to replicate our experimental environment and reproduce the findings.

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

## A  ERROR ANALYSIS ON THE EXAMPLE TASK

```
Image:           8
         6            1

      7                  3

         4            2
                5

Function Signature:
from typing import Tuple

def solution(line1: Tuple[int, int], line2: Tuple[int, int]) -> bool:
    """
    Determines if two line segments ultimately intersect.

    Parameters:
    line1: A tuple containing the numbers at both ends of the first line segment.
    line2: A tuple containing the numbers at both ends of the second line segment.

    Returns:
    bool: True if the line segments intersect, False otherwise.
    """
```

```
An Incorrect Code Solution Generated by GPT-4o:
def solution(line1: Tuple[int, int], line2: Tuple[int, int]) -> bool:
    def is_between(a, b, c):
        if a < b:
            return a < c < b
        return a < c or c < b
    a, b = line1
    c, d = line2
    if a > b:
        a, b = (b, a)
    if c > d:
        c, d = (d, c)
    return is_between(a, b, c) != is_between(a, b, d) and is_between(c, d, a) != is_between(c, d, b)
```

```
An Incorrect Code Solution Generated by Claude 3.5 Sonnet:
def solution(line1: Tuple[int, int], line2: Tuple[int, int]) -> bool:
    def normalize_segment(segment):
        return (min(segment), max(segment))
    line1 = normalize_segment(line1)
    line2 = normalize_segment(line2)
    if line1[0] <= line2[0] and line1[1] >= line2[1] or (line2[0] <= line1[0] and line2[1] >= line1[1]):
        return True
    if max(line1[0], line2[0]) <= min(line1[1], line2[1]):
        return True
    if line1[1] < line2[0] and line2[1] < line1[0] or (line2[1] < line1[0] and line1[1] < line2[0]):
        return True
    return False
```

Figure 5: Examples of incorrect solutions generated by GPT-4o and Claude 3.5 Sonnet for the coding task shown in Figure 1.

Figure 1 illustrates a simple coding task in `HumanEval-V`. The task requires determining whether two line segments, defined by pairs of numbers on a clock-like circle, will ultimately intersect if allowed to extend outside the circle. The numbers on the circle are arranged in a non-standard order. Despite the problem's simplicity, all evaluated LMMs failed to solve it correctly even when generating 20 samples. We present representative solutions generated by GPT-4o and Claude 3.5 Sonnet in Figure 5.

Both models implement sorting-based algorithms that compare the numbers at the endpoints of the line segments. However, they fail to account for the critical scenario where the segments intersect outside the circle, and fail to recognize the unordered arrangement of the numbers. This oversight indicates that the models are not effectively capturing the essential visual details of the problem. Notably, this issue appears to stem from data leakage, as the original coding task is derived from a CodeForces problem (https://codeforces.com/contest/1971/problem/C), and the generated solutions in Figure 5 reflect patterns more suitable for the original context. This phenomenon is not isolated to this task; we observe similar issues across many coding tasks in `HumanEval-V`. This highlights that the models rely on memorized patterns instead of genuinely understanding the visual context. Such failures emphasize the importance of preventing data leakage and validate the rationale behind our careful adaptation and mutation processes during data annotation.

| LMMs | Params | pass@1 | pass@10 | pass@$k$ ($n = 100$) | | | |
| --- | --- | --- | --- | --- | --- | --- | --- |
| | | | | $k$=10 | $k$=20 | $k$=50 | $k$=100 |
| Proprietary | | | | | | | |
| GPT-4o | | **13.0** | **36.4** | **39.0** | **44.1** | **49.9** | **53.7** |
| GPT-4o-mini | | 6.5 | 15.4 | 15.3 | 20.1 | 26.7 | 31.5 |
| Open-Weight | | | | | | | |
| InternVL-2 | 40.1B | 0.0 | 1.6 | 3.0 | 4.9 | 8.0 | 10.2 |
| InternVL-2 | 25.5B | 0.0 | **3.2** | **3.2** | 4.9 | 7.7 | 10.2 |
| InternVL-2 | 8.1B | **0.9** | 2.6 | 3.0 | 5.0 | 8.4 | 10.2 |
| InternVL-2 | 4.2B | 0.0 | 2.3 | 2.3 | 4.4 | **9.4** | **14.8** |
| Qwen2-VL | 8.3B | 0.0 | 1.6 | 3.1 | 5.2 | 8.7 | 11.1 |
| LLaVA-OneVision | 8.0B | **0.9** | 1.9 | 1.9 | 3.4 | 6.7 | 10.2 |
| InternVL-Chat-V1.5 | 25.5B | 0.0 | 2.1 | 3.1 | **5.3** | 9.3 | 13.0 |
| MiniCPM-V 2.6 | 8.1B | **0.9** | 2.2 | 1.7 | 2.8 | 4.8 | 7.4 |
| MiniCPM-V 2.5 | 8.5B | 0.0 | 2.3 | 1.3 | 2.4 | 5.5 | 9.3 |
| Phi-3.5-Vision | 4.2B | **0.9** | 1.6 | 2.1 | 3.3 | 5.0 | 6.5 |
| Phi-3-Vision | 4.2B | 0.0 | 2.6 | 1.8 | 3.3 | 6.6 | 9.3 |

Table 5: The performance of 13 LMMs on `HumanEval-V` with more generated code solution samples. The pass@1 and pass@10 columns are the results from Table 2. Scores are shown as percentages, with the highest values highlighted in **bold**.

| LMMs | pass@1 | pass@10 | pass@$k$ ($n = 1,000$) | | | | | |
| --- | --- | --- | --- | --- | --- | --- | --- | --- |
| | | | $k$=100 | $k$=200 | $k$=400 | $k$=600 | $k$=800 | $k$=1000 |
| GPT-4o | 13.0 | 36.4 | 55.3 | 59.9 | 64.3 | 66.4 | 67.7 | 68.5 |
| GPT-4o-mini | 6.5 | 15.4 | 31.3 | 36.0 | 40.5 | 43.0 | 44.9 | 46.3 |

Table 6: The impact of scaling the number of samples on `HumanEval-V`. Scores are reported as percentages. The pass@1 and pass@10 columns correspond to results from Table 2.

# B ADDITIONAL EXPERIMENTAL RESULTS

## B.1 PERFORMANCE WITH MORE SAMPLES

The results in Section 4.1 indicate that increasing the number of samples can significantly enhance model performance on `HumanEval-V`, so we conduct an ablation study to examine the effect of scaling up sample sizes. Due to budgetary constraints, we primarily test open-weight LMMs ranging from 4B to 40B parameters. For proprietary models, we evaluate GPT-4o and GPT-4o-mini. For all selected models, we increase the number of generated samples $n$ to 100 to observe their performance. The results are presented in Table 5.

From the results, we observe that increasing the sample size consistently improves performance across most models. For example, GPT-4o achieves a substantial improvement, rising from 36.4% pass@10 to 53.7% pass@100. Smaller-scale open-weight LMMs also show notable gains; for instance, InternVL-2 (4.2B) improves from a pass@10 of 2.3% to a pass@100 of 14.8%. However, not all models benefit equally from scaling the sample size. For instance, Phi-3.5-Vision, which has the same 4B-level parameter size, achieves only a pass@100 score of 6.5%. These findings underscore both the potential and the limitations of scaling sample numbers to improve current LMMs' performance on `HumanEval-V`.

To further evaluate the performance of current LMMs, we increase the sample size for GPT-4o to 1,000. The results, presented in Table 6, show promising results with GPT-4o achieving a pass@1000 of 68.5%, compared to the 36.4% pass@10. Similarly, GPT-4o-mini demonstrates strong performance, achieving a pass@1000 score of 46.3%, surpassing the pass@10 score of GPT-4o. These findings suggest that a significant proportion of the coding tasks in `HumanEval-V` are solvable with current LMM capabilities, highlighting the need for further research on strategies to better motivate the abilities of LMMs.

It is important to note that there may be some variance between the pass@10 scores reported with $n$=20 and those with $n$=100 or $n$=1,000. Increasing $n$ typically improves the accuracy of the estimated pass@$k$, making comparisons between different $n$ values less straightforward. Moreover, the pass@100 and pass@1000 values reported in Table 5 and Table 6 may exhibit bias due to using the same $k$ and $n$ values for calculating pass@$k$, potentially affecting reproducing the results.

## B.2 COMPARISON WITH OTHER MULTIMODAL BENCHMARKS

| Models | Params | Multidisciplinary Multimodal Benchmarks | | | | | HumanEval-V | |
| | | MMMU | MathVista | MMVet | MME | RealWorldQA | pass@1 | pass@10 |
|---|---|---|---|---|---|---|---|---|
| *Proprietary* | | | | | | | | |
| GPT-4o | | 69.2 | 61.3 | 69.1 | 2310.3 | 75.4 | 13.0 | 36.4 |
| GPT-4o-mini | | 60.0 | 52.4 | 66.9 | 2003.4 | 67.1 | 6.5 | 15.4 |
| Claude 3.5 Sonnet | | 65.9 | 61.6 | 66.0 | 1920.0 | 60.1 | 18.5 | 25.9 |
| Gemini 1.5 Pro | | 60.6 | 57.7 | 64.0 | 2110.6 | 64.1 | 10.2 | 22.2 |
| Gemini 1.5 Flash | | 58.2 | 51.2 | 63.2 | 2077.9 | 69.0 | 8.3 | 13.2 |
| *Open-Weight* | | | | | | | | |
| InternVL-2 | 76.3B | 58.3 | 65.6 | 64.4 | 2397.6 | 72.7 | 3.7 | 12.8 |
| | 40.1B | 55.2 | 64.0 | 61.8 | 2293.1 | 70.1 | 0.0 | 1.6 |
| | 25.5B | 50.7 | 59.4 | 60.0 | 2259.8 | 68.1 | 0.0 | 3.2 |
| | 8.1B | 51.2 | 58.3 | 54.3 | 2215.1 | 64.2 | 0.9 | 2.6 |
| | 4.2B | 48.3 | 58.1 | 50.9 | 2064.6 | 60.5 | 0.0 | 2.3 |
| Qwen2-VL | 73.4B | 64.5 | 70.5 | 74.0 | 2482.7 | 77.8 | 3.7 | 16.0 |
| | 8.3B | 54.1 | 58.2 | 62.0 | 2326.8 | 70.1 | 0.0 | 1.6 |
| LLaVA-OneVision | 73.2B | 56.8 | 67.5 | 63.7 | 2261.0 | 71.9 | 1.9 | 12.4 |
| | 8.0B | 48.8 | 63.2 | 57.5 | 1998.0 | 66.3 | 0.9 | 1.9 |
| InternVL-Chat-V1.5 | 25.5B | 46.8 | 54.7 | 55.4 | 2189.6 | 65.6 | 0.0 | 2.1 |
| MiniCPM-V 2.6 | 8.1B | 49.8 | 60.6 | 60.0 | 2268.7 | 65.0 | 0.9 | 2.2 |
| MiniCPM-V 2.5 | 8.5B | 45.8 | 54.3 | 52.8 | 2024.6 | 63.5 | 0.0 | 2.3 |
| Phi-3.5-Vision | 4.2B | 44.6 | 43.2 | 43.2 | 1838.1 | 53.6 | 0.9 | 1.6 |
| Phi-3-Vision | 4.2B | 46.1 | 44.6 | 44.1 | 1508.0 | 58.8 | 0.0 | 2.6 |

Table 7: A performance comparison of 19 LMMs on HumanEval-V and five other popular multimodal benchmarks. The pass@1 and pass@10 columns correspond to results from Table 2. Values are highlighted using a blue color scale, where darker shades indicate higher scores.

| | MMMU | MathVista | MMVet | MME | RealWorldQA | HumanEval-V |
|---|---|---|---|---|---|---|
| MMMU | - | 0.51 | 0.88 | 0.42 | 0.61 | 0.90 |
| MathVista | 0.51 | - | 0.72 | 0.77 | 0.73 | 0.28 |
| MMVet | 0.88 | 0.72 | - | 0.68 | 0.81 | 0.67 |
| MME | 0.42 | 0.77 | 0.68 | - | 0.80 | 0.17 |
| RealWorldQA | 0.61 | 0.73 | 0.81 | 0.80 | - | 0.38 |
| HumanEval-V | 0.90 | 0.28 | 0.67 | 0.17 | 0.38 | - |
| Average | 0.66 | 0.60 | 0.75 | 0.57 | 0.67 | 0.48 |

Table 8: The Pearson correlation coefficients between pairs of six multimodal benchmarks. Lower correlation values highlight benchmarks that capture distinct aspects of model performance.

To analyze whether HumanEval-V identifies specific weaknesses that are not captured by existing benchmarks, we select five widely used multimodal benchmarks that cover multidisciplinary abilities. The selected benchmarks include MMMU (Yue et al., 2024), MathVista (Lu et al., 2023), MMVet (Yu et al., 2023), MME (Fu et al., 2023), and RealWorldQA (xAI, 2024). We collect the performance results of the 19 LMMs evaluated in this paper from the OpenVLM Leaderboard (Duan et al., 2024) and the corresponding papers and reports. These results are presented alongside the pass@1 and pass@10 scores on HumanEval-V in Table 7. From the results, we observe that open-weight LMMs with over 70B parameters generally perform well on the selected benchmarks, with models such as InternVL-2 (76.3B) and Qwen2-VL (73.4B) even surpassing proprietary models

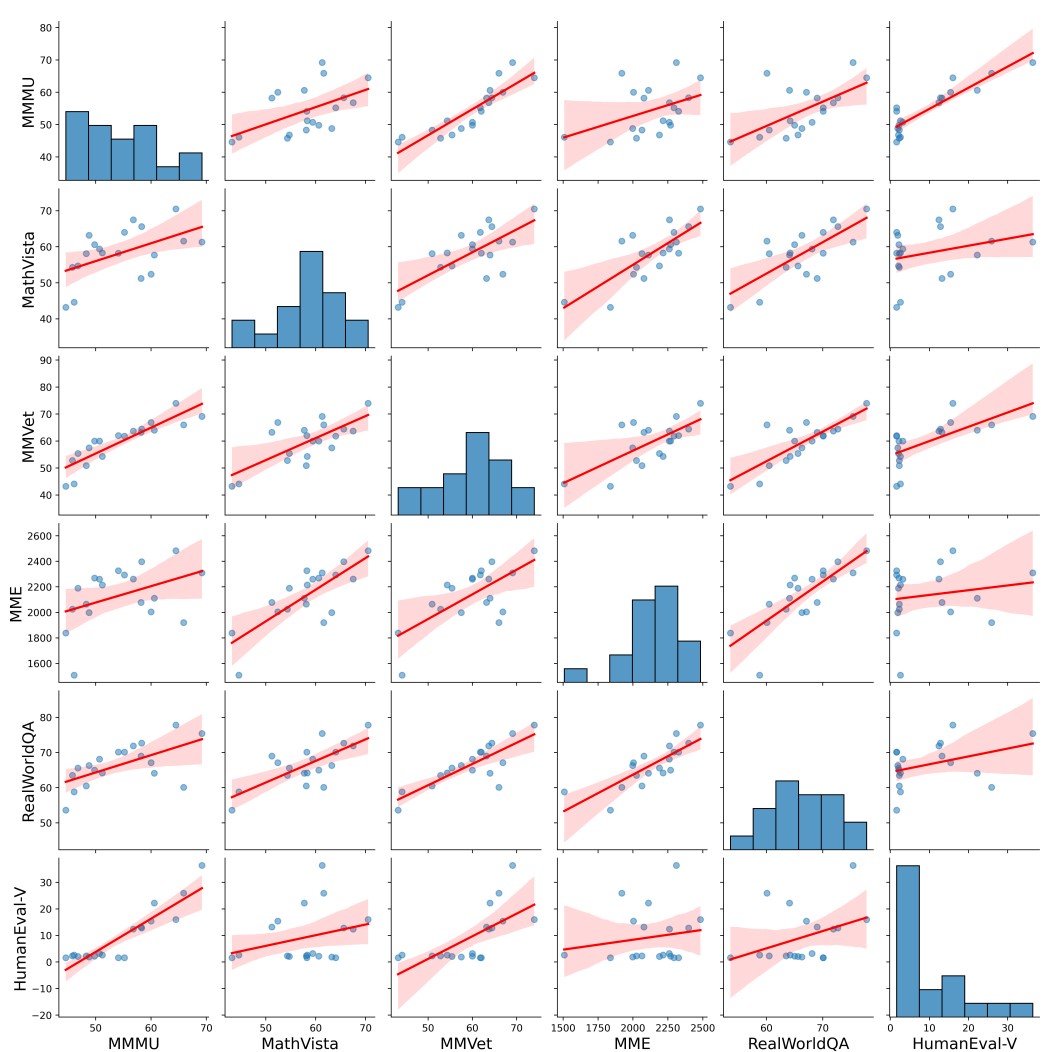

Figure 6: The correlations between six multimodal benchmarks, including `HumanEval-V`. Each subplot, except on the diagonal, visualizes the relationship between two benchmarks.

like GPT-4o and Claude 3.5 Sonnet in some cases. However, these open-weight LMMs show significantly lower performance on `HumanEval-V`, indicating that our benchmark can uncover model weaknesses that are not apparent in other evaluations.

To quantify the relationship between `HumanEval-V` and the five selected benchmarks, we calculate the Pearson correlation coefficient using the data in Table 7. The results, shown in Table 8, reveal that `HumanEval-V` has the lowest average correlation coefficient across all benchmarks, suggesting that it captures aspects of model performance that are overlooked by existing benchmarks. Among the benchmarks, `HumanEval-V` shows the highest correlation with MMMU, which primarily evaluates advanced perception and reasoning abilities—key focuses of our benchmark as well. We also visualize these relationships using regression plots for each benchmark pair in Figure 6, providing an intuitive view of the correlations. From the plots, we observe that many of the scatter points for `HumanEval-V` are concentrated around zero, contributing to the low correlation with other benchmarks and highlighting the distinct challenges posed by our benchmark.

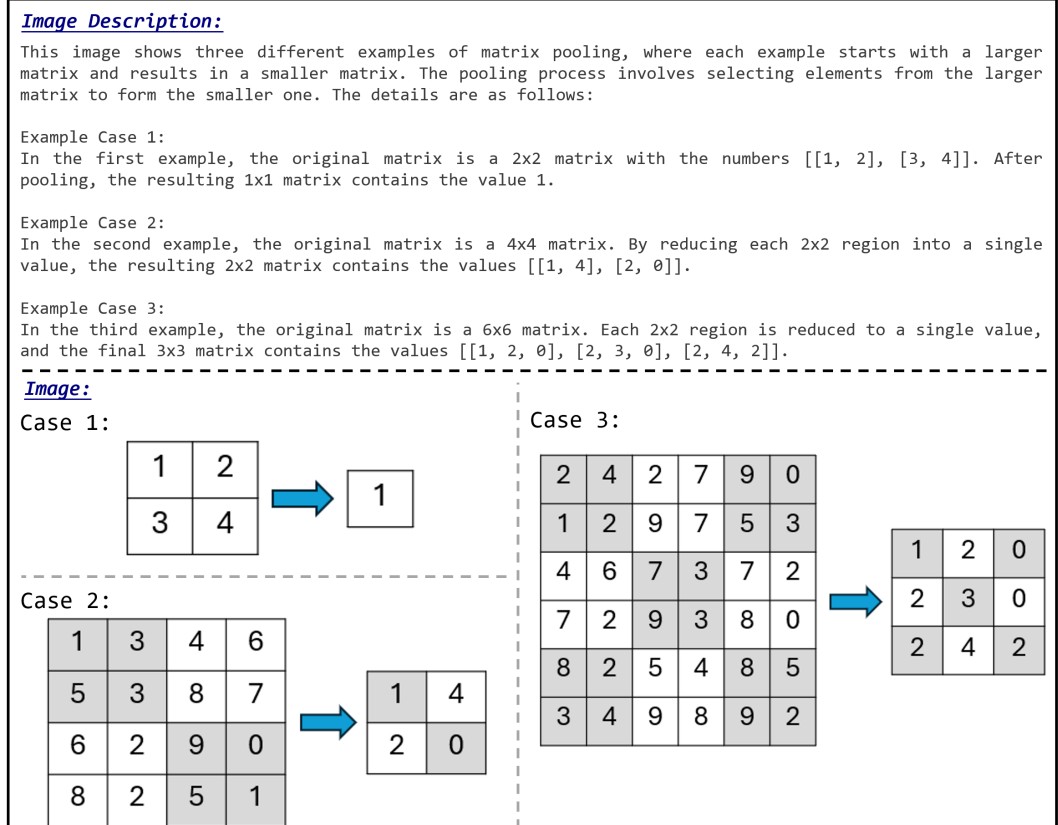

```
Image Description:

This image shows three different examples of matrix pooling, where each example starts with a larger
matrix and results in a smaller matrix. The pooling process involves selecting elements from the larger
matrix to form the smaller one. The details are as follows:

Example Case 1:
In the first example, the original matrix is a 2x2 matrix with the numbers [[1, 2], [3, 4]]. After
pooling, the resulting 1x1 matrix contains the value 1.

Example Case 2:
In the second example, the original matrix is a 4x4 matrix. By reducing each 2x2 region into a single
value, the resulting 2x2 matrix contains the values [[1, 4], [2, 0]].

Example Case 3:
In the third example, the original matrix is a 6x6 matrix. Each 2x2 region is reduced to a single value,
and the final 3x3 matrix contains the values [[1, 2, 0], [2, 3, 0], [2, 4, 2]].
```

Figure 7: An example of image description annotation.

### B.3 EXPERIMENTING WITH IMAGE DESCRIPTIONS

We provide two examples in Figure 7 and Figure 8 to illustrate our annotation process and demonstrate how we construct image descriptions. When creating these descriptions, we ensure they are purely descriptive rather than instructive, refraining from disclosing any specific algorithms or problem-solving strategies. This approach allows us to evaluate whether current LMMs possess genuine visual understanding capabilities and whether they can perform well when the visual elements are grounded through detailed textual descriptions.

This process poses a unique challenge. While humans can intuitively identify patterns in images and summarize them succinctly, we require our annotators to use precise descriptive language that details every visual aspect without inferring the specific steps to solve the problem. This increases the complexity of annotation and often results in verbose image descriptions. Despite this verbosity, maintaining a purely descriptive approach is crucial for our benchmark, as it ensures that solving the task requires the model to interpret and reason about the visual content, rather than simply interpreting the description into code.

Once the image descriptions are finalized, we employ the prompt template shown in Figure 9 to guide the LMMs in generating code solutions for the tasks in `HumanEval-V`.

## C BENCHMARK CONSTRUCTION DETAILS

### C.1 ADDITIONAL DETAILS OF DATA COLLECTION

Our data collection process involves two primary sources: Stack Overflow (SO) and coding challenge platforms. Each coding problem undergoes a strict screening process to ensure that it aligns with the standards of `HumanEval-V`. Annotators are instructed to identify suitable problems by

```
Image Description:
The image demonstrates the logical **AND** operation between segments of seven-segment displays, and in
each result, certain segments are **highlighted in red** to indicate the active segments common to both
input digits.

Here's a detailed description of each example with the highlighted segments:

### **First Example**:
- **Operation**: **0 & 1**
  - **Result**: Digit **1**, where only the two right segments are **red**.

### **Second Example**:
- **Operation**: **5 & 3**
  - **Result**: The top, middle, right-bottom vertical and bottom horizontal segments are highlighted in
**red**.

### **Third Example**:
- **Operation**: **2 & 6 & 4**
  - **Result**: Only middle horizontal segment is **red**

### **Fourth Example**:
- **Operation**: **7 & 8 & 9**
  - **Result**: Digit **7** in **red**.
```

Image:

Figure 8: An example of image description annotation.

```
**Instructions:**
You are an exceptionally intelligent coding assistant that consistently delivers accurate and reliable
responses to user instructions. Please complete the function based on the provided image with textual
description and the code context. Return the complete solution, including the function signature, in a
single response, formatted within a Python code block.

**Image Description:**
```
{image_description}
```

**Code Context:**
```python
{code_context}
```
```

Figure 9: The template used for prompting LMMs to solve code generation tasks with image descriptions. The {image_description} placeholder is replaced with the annotated image description. The {code_context} placeholder is replaced with the corresponding function signature.

assessing whether they can be adapted with minimal effort to meet the predefined standards, which include the following criteria: (1) the visual context must be essential to solving the task, with all relevant visual information able to fit within a single image; (2) the problem should be largely self-explanatory through its visual context, requiring minimal textual description; and (3) the problem should target entry-level programmers and be solvable using only common Python libraries.

We select SO due to its extensive repository of real-world programming problems. To identify relevant posts, we filter for questions from 2020 that have non-negative votes and accepted answers. Next, we focus on posts with images in the question body and code blocks in the corresponding answers, narrowing down to those tagged with python. After this automated filtering, we manually

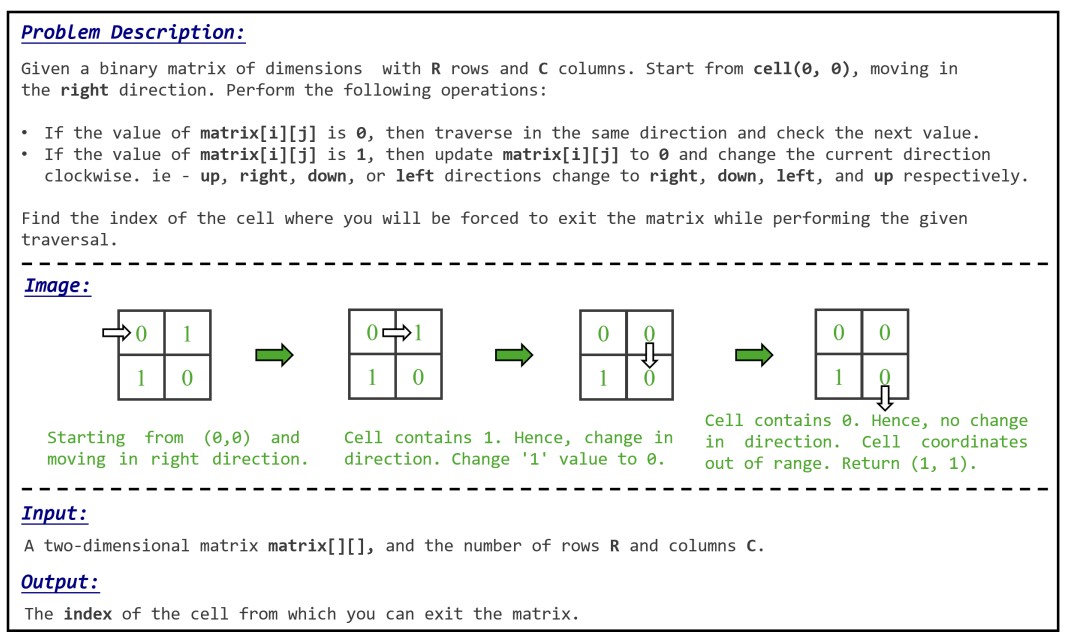

---

**Problem Description:**

Shaass has $n$ books. He wants to make a bookshelf for all his books. He wants the bookshelf's dimensions to be as small as possible. The **thickness** of the $i$-th book is $t_i$ and its pages' **width** is equal to $w_i$. The thickness of each book is either **1** or **2**. All books have the **same page heights**.

Shaass puts the books on the bookshelf in the following way. First he selects some of the books and put them vertically. Then he puts the rest of the books **horizontally above the vertical books**. The sum of the widths of the horizontal books must be **no more than** the total thickness of the vertical books. A sample arrangement of the books is depicted in the image.

Help Shaass to find the **minimum total thickness** of the vertical books that we can achieve.

**Image:**

**Input:**

The first line of the input contains an integer $n$, ($1 \leq n \leq 100$). Each of the next $n$ lines contains two integers $t_i$ and $w_i$ denoting the **thickness** and **width** of the $i$-th book correspondingly, ($1 \leq t_i \leq 2$, $1 \leq w_i \leq 100$).

**Output:**

On the only line of the output print the **minimum total thickness** of the vertical books that we can achieve.

---

Figure 10: A negative example in our data screening process, sourced from CodeForces (`https://codeforces.com/problemset/problem/294/B`), where the image is non-essential for solving the problem.

---

**Problem Description:**

Given a binary matrix of dimensions with **R** rows and **C** columns. Start from **cell(0, 0)**, moving in the **right** direction. Perform the following operations:

- If the value of **matrix[i][j]** is **0**, then traverse in the same direction and check the next value.
- If the value of **matrix[i][j]** is **1**, then update **matrix[i][j]** to **0** and change the current direction clockwise. ie - **up**, **right**, **down**, or **left** directions change to **right**, **down**, **left**, and **up** respectively.

Find the index of the cell where you will be forced to exit the matrix while performing the given traversal.

**Image:**

Starting from (0,0) and moving in right direction.

Cell contains 1. Hence, change in direction. Change '1' value to 0.

Cell contains 0. Hence, no change in direction. Cell coordinates out of range. Return (1, 1).

**Input:**

A two-dimensional matrix **matrix[][],** and the number of rows **R** and columns **C**.

**Output:**

The **index** of the cell from which you can exit the matrix.

---

Figure 11: A negative example in our data screening process, sourced from GeeksforGeeks (`https://www.geeksforgeeks.org/problems/last-cell-in-a-matrix/1`), where the visual elements require extensive textual descriptions to interpret.

review the remaining posts, excluding topics such as front-end, mobile, or UI development, as these often require high-level API usage and do not align with the goals of our benchmark. We also filter out many posts containing images that only provide supplementary details (e.g., code snippets, error messages, or execution outputs) rather than being essential to problem-solving. Ultimately,

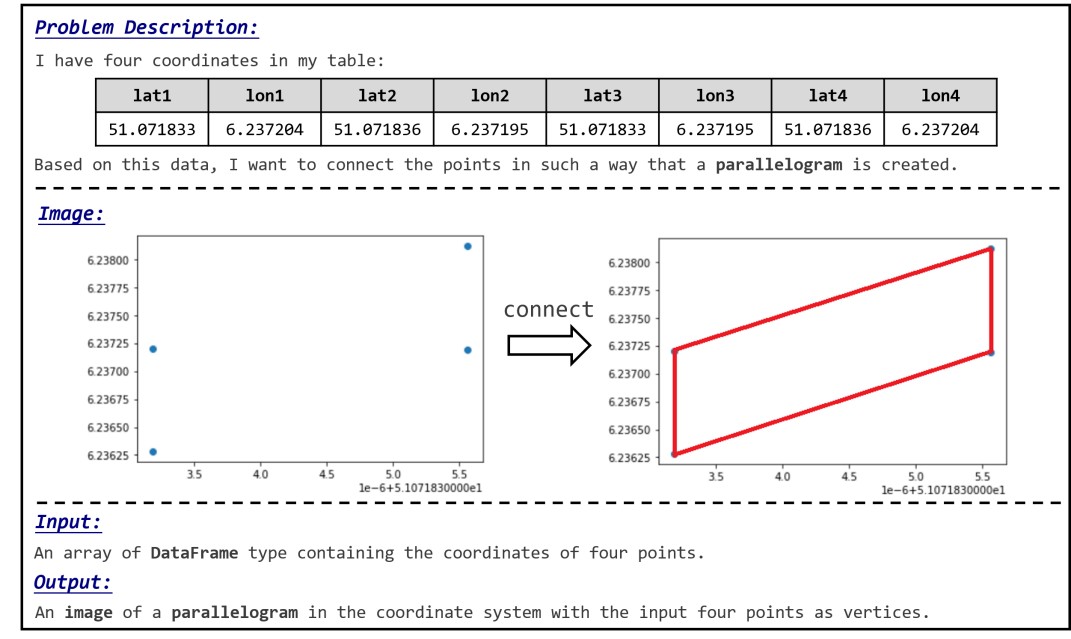

Figure 12: A positive example in our data screening process, sourced from Stack Overflow (`https://stackoverflow.com/questions/69163515`).

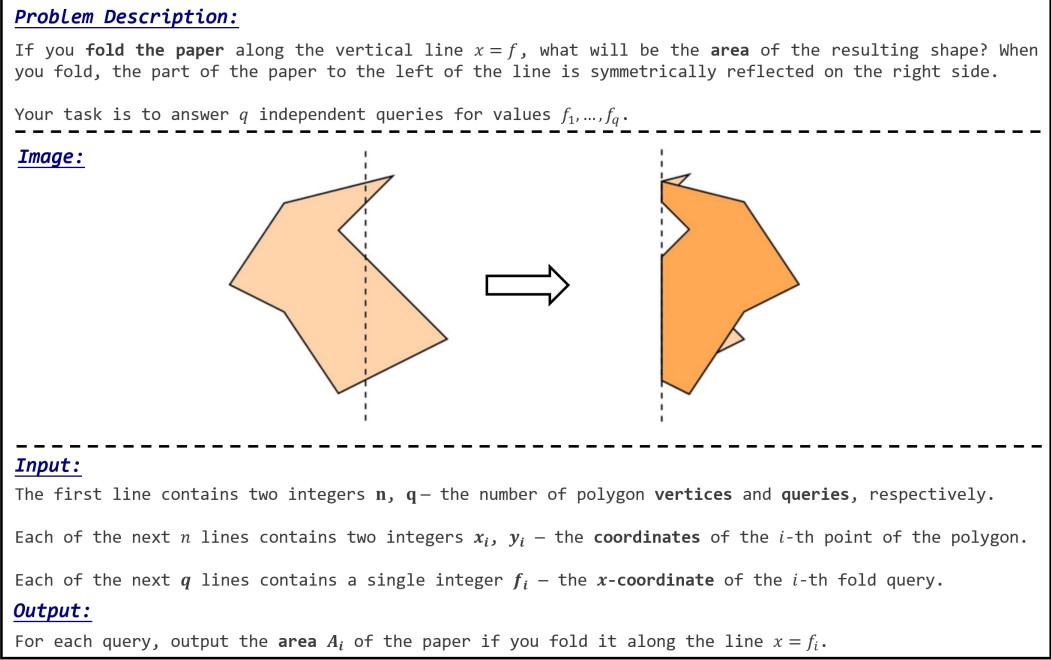

Figure 13: A positive example in our data screening process, sourced from CodeForces (`https://codeforces.com/problemset/problem/1381/E`).

we identify 8 posts satisfying our standards, covering topics like geometry, plotting, and image processing. The final screened SO posts account for less than 1% of the total viewed posts, and even the selected problems often require significant adaptation to fit our benchmark's requirements.

Regarding the coding challenge platforms, we utilize the open-source MMCode dataset Li et al. (2024b), which already scraped coding problems from various coding challenge platforms that in-

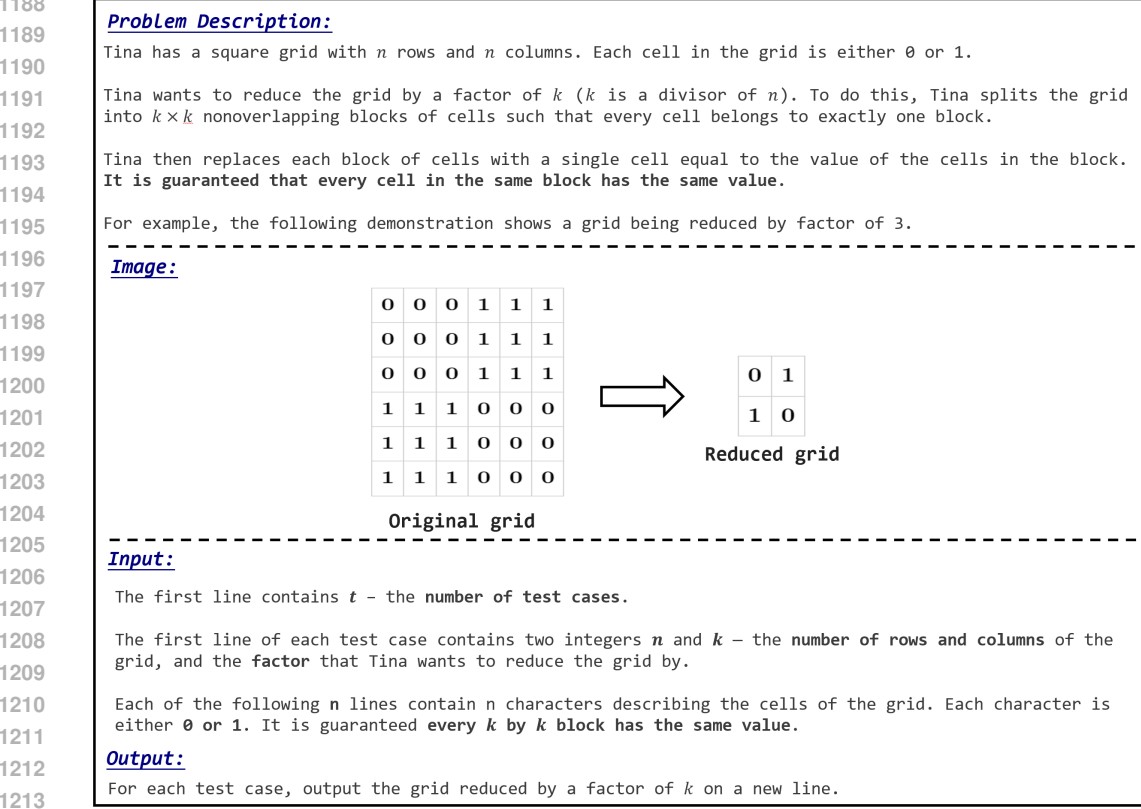

*Problem Description:*

Tina has a square grid with $n$ rows and $n$ columns. Each cell in the grid is either 0 or 1.

Tina wants to reduce the grid by a factor of $k$ ($k$ is a divisor of $n$). To do this, Tina splits the grid into $k \times k$ nonoverlapping blocks of cells such that every cell belongs to exactly one block.

Tina then replaces each block of cells with a single cell equal to the value of the cells in the block. **It is guaranteed that every cell in the same block has the same value.**

For example, the following demonstration shows a grid being reduced by factor of 3.

*Image:*

*Input:*

The first line contains $t$ – the **number of test cases**.

The first line of each test case contains two integers $n$ and $k$ – the **number of rows and columns** of the grid, and the **factor** that Tina wants to reduce the grid by.

Each of the following $n$ lines contain n characters describing the cells of the grid. Each character is either **0** or **1**. It is guaranteed **every $k$ by $k$ block has the same value.**

*Output:*

For each test case, output the grid reduced by a factor of $k$ on a new line.

Figure 14: A positive example in our data screening process, sourced from CodeForces (`https://codeforces.com/problemset/problem/1996/B`).

corporate visual elements in problem descriptions. However, we find that most of these problems are unsuitable for `HumanEval-V`. Many images merely display simple mathematical equations, which are essentially textual in nature and do not require visual reasoning. In other cases, the visual content is redundant, as it can be easily inferred from the text alone, rendering the images non-essential. Some problems, although containing relevant visual information, are overly complex and require extensive textual descriptions to interpret, violating our requirement for self-explanatory visual contexts. After careful screening, we identify 32 problems suitable for our benchmark: 23 from CodeForces, 5 from LeetCode, and 1 each from GeeksforGeeks, AtCoder, Open Kattis, and Project Euler. These selected problems account for less than 5% of the total viewed problems.

To further illustrate our screening process, we present two negative examples that do not meet our standards in Figure 10 and Figure 11, along with three positive examples that are selected for our benchmark in Figure 12, Figure 13, and Figure 14. Below are the detailed explanations:

In Figures 10 and 11, we present two negative examples that do not meet the standards for inclusion in our benchmark. Figure 10 is a coding problem sourced from CodeForces that requires determining an optimal stacking method for a set of books with identical heights, given their respective thickness and width, to minimize the total thickness. Although the provided image illustrates a possible stacking configuration, it lacks essential information, such as constraints on the stacking method and precise book dimensions. Furthermore, the core problem-solving information is conveyed predominantly through text, making the image non-essential for understanding the solution. Figure 11 depicts a coding problem from GeeksForGeeks that involves traversing a 2D matrix according to a specified pattern, starting from the top-left corner and identifying the traversal endpoint. Although the image provides a basic representation of the matrix, the traversal pattern is too intricate to be effectively captured visually and requires substantial textual explanation. As a result, the textual description contains more problem-solving information than the image itself, violating our requirement that the visual context be self-explanatory and the primary source of information.

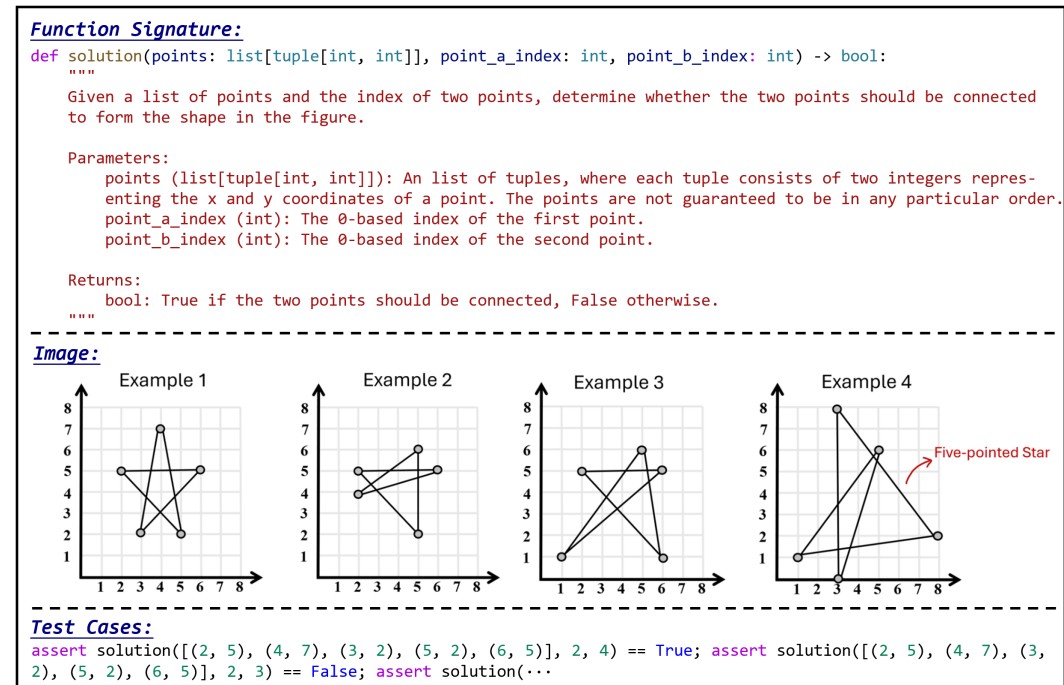

Figure 15: The adapted coding task from Figure 12 as incorporated into `HumanEval-V`.

In Figure 12, Figure 13, and Figure 14, we present three examples that are well-suited for inclusion in our benchmark. Figure 12 illustrates a practical problem from Stack Overflow, where a developer seeks to draw a parallelogram on a coordinate plane using four specified points. The image visually demonstrates how these points are connected to form the parallelogram, serving as the critical piece of information needed to solve the task. Additionally, the text merely reiterates the geometric properties shown in the image, making it possible to reduce the textual content significantly without loss of essential details. This ensures that the image itself is indispensable for solving the problem while relying on the text alone would be insufficient. Figure 13 features a problem from CodeForces involving the folding of a polygon, where the goal is to compute the area of the resulting shape after a series of folds. The image clearly depicts the folding process along the designated dashed lines, showing both the original shape and its transformation after folding. These visual details are integral to solving the problem, as understanding the fold pattern and resulting shape is necessary. Figure 14, also sourced from CodeForces, involves reducing a grid according to a specified pattern. The image effectively conveys the grid reduction process, showing the transformation step-by-step. Any redundant textual description of the pattern can be omitted, ensuring that the problem can be solved primarily by interpreting the visual information, with minimal reliance on the accompanying text. These three examples are relatively straightforward yet require precise visual understanding, making them ideal candidates for adaptation into coding tasks within `HumanEval-V`.

## C.2 EXAMPLES OF ADAPTING CODING PROBLEMS

We present three adapted examples in Figure 15, Figure 16, and Figure 17, derived from the original coding tasks shown in Figure 12, Figure 13, and Figure 14. For each problem, we redesign the questions, redraw the accompanying images to include the critical problem-solving context, and simplify the textual descriptions. Furthermore, we adjust the difficulty to ensure that entry-level programmers can interpret the visual information accurately and implement the solution using basic coding skills.

In Figure 15, we transform the original parallelogram problem into the coding task involving a five-pointed star, incorporating richer visual information. To enhance the visual cues, we include four examples in the image demonstrating different ways to connect five points to form a star. In the adapted function signature, we specify the implementation requirements for the model, clearly

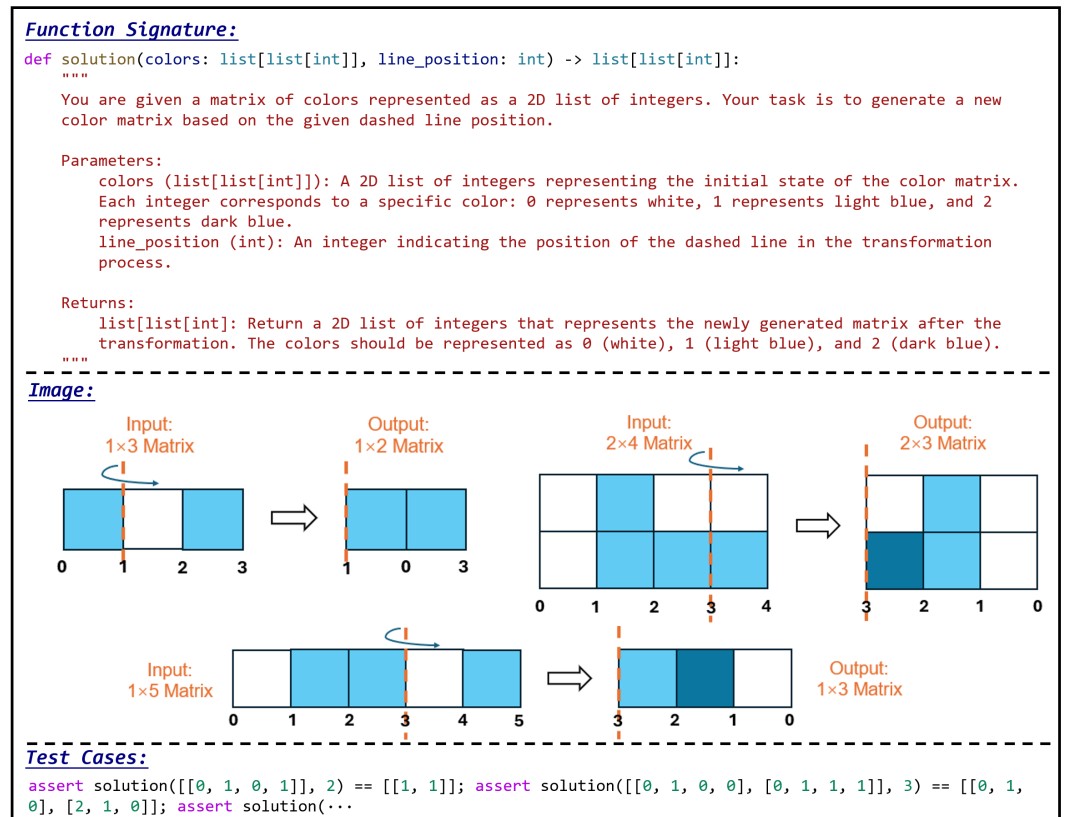

```
Function Signature:
def solution(colors: list[list[int]], line_position: int) -> list[list[int]]:
    """
    You are given a matrix of colors represented as a 2D list of integers. Your task is to generate a new
    color matrix based on the given dashed line position.

    Parameters:
        colors (list[list[int]]): A 2D list of integers representing the initial state of the color matrix.
        Each integer corresponds to a specific color: 0 represents white, 1 represents light blue, and 2
        represents dark blue.
        line_position (int): An integer indicating the position of the dashed line in the transformation
        process.

    Returns:
        list[list[int]]: Return a 2D list of integers that represents the newly generated matrix after the
        transformation. The colors should be represented as 0 (white), 1 (light blue), and 2 (dark blue).
    """
```

*Image:*

*Test Cases:*

```
assert solution([[0, 1, 0, 1]], 2) == [[1, 1]]; assert solution([[0, 1, 0, 0], [0, 1, 1, 1]], 3) == [[0, 1,
0], [2, 1, 0]]; assert solution(···
```

Figure 16: The adapted coding task from Figure 13 as incorporated into `HumanEval-V`.

defining the function's objectives, input parameters, and constraints on the return value. Unlike the original problem, which requires generating an image of a parallelogram, the adapted task simply asks whether two specified points should be connected. This adaptation reduces the complexity while maintaining a strong focus on assessing the model's visual reasoning abilities. Additionally, the structured I/O format allows us to evaluate the generated solutions through test cases.

In Figure 16, we simplify the original polygon folding problem into a matrix folding task. After folding, overlapping sections of the matrix result in color changes, and the model is required to determine the resulting color distribution. We restrict the input matrix to two initial colors: white and light blue, such that after folding, the matrix can display three distinct color outcomes: white, light blue, and dark blue. This adaptation preserves the visual reasoning involved in understanding the folding process while reducing the programming difficulty. We also provide three illustrative examples within the image to ensure clarity.

In Figure 17, we slightly increase the difficulty of the original problem. We remove redundant textual details that can be inferred from the image. We omit the reduction factor $k$ from the function parameters, setting $k$ as a fixed value instead. The model is expected to deduce that $k = 2$ based on the three provided examples. Moreover, instead of performing simple scaling operations with 0 and 1 values as in the original problem, we adapt it into a pooling operation based on statistical features (e.g., determining the minimum value), which requires not only OCR capabilities but also deeper visual reasoning.

### C.3 EXAMPLES OF MUTATING CODING TASKS

We apply mutations to some of the 40 screened coding tasks to expand the volume of our benchmark. The objective is to generate new tasks that retain the essence of the original tasks but introduce distinct patterns with minimal modification. As illustrated in Figures 18, Figure 19, and Figure 20, these mutated tasks are derived from the coding problems in Figures 15, 16, and 17, respectively.

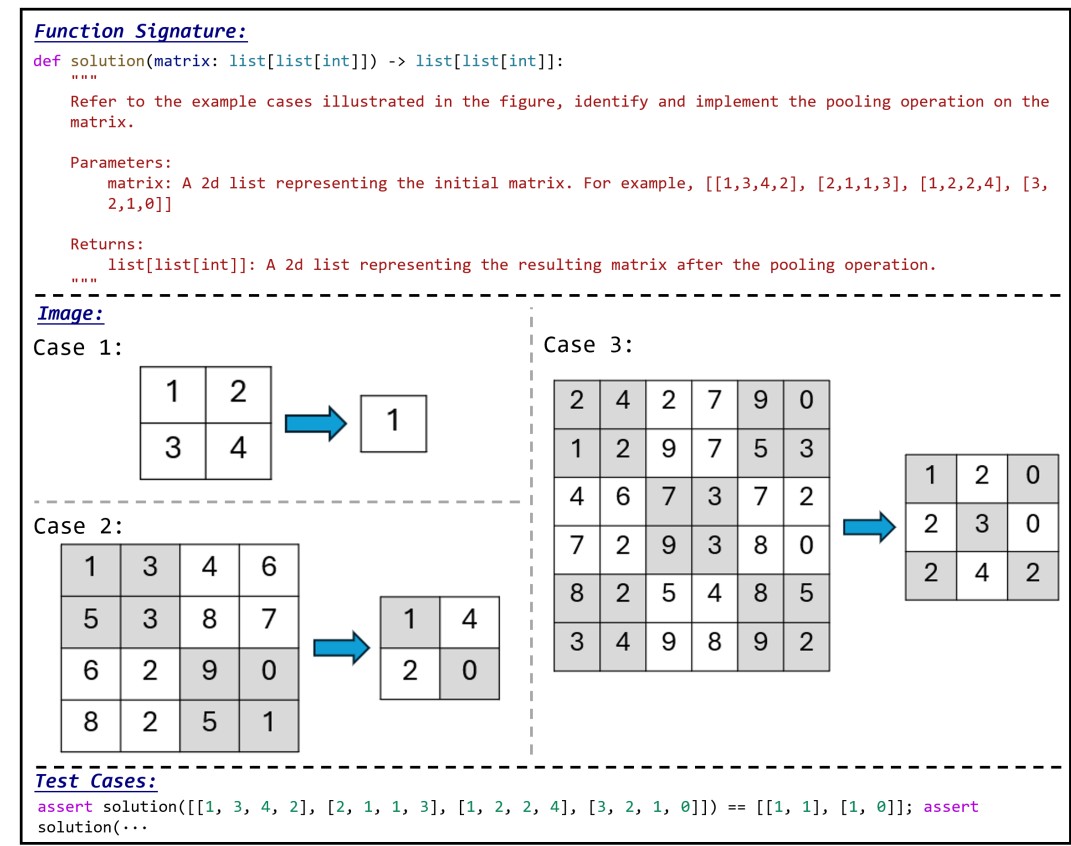

```
Function Signature:
def solution(matrix: list[list[int]]) -> list[list[int]]:
    """
    Refer to the example cases illustrated in the figure, identify and implement the pooling operation on the
    matrix.

    Parameters:
        matrix: A 2d list representing the initial matrix. For example, [[1,3,4,2], [2,1,1,3], [1,2,2,4], [3,
        2,1,0]]

    Returns:
        list[list[int]]: A 2d list representing the resulting matrix after the pooling operation.
    """
```

Image:

Case 1:

Case 2:

Case 3:

```
Test Cases:
assert solution([[1, 3, 4, 2], [2, 1, 1, 3], [1, 2, 2, 4], [3, 2, 1, 0]]) == [[1, 1], [1, 0]]; assert
solution(···
```

Figure 17: The adapted coding task from Figure 14 as incorporated into `HumanEval-V`.

In Figure 18, we maintain the same function signature as in the original task but modify the image pattern from a five-pointed star to a six-pointed star, altering the visual configuration while preserving the overall task settings. In Figure 19, we transform the color addition rule in the folded matrix into a numeric addition rule, requiring the model to recognize and infer the numerical changes before and after folding. This mutation introduces additional complexity, further evaluating the model's OCR capabilities. For Figure 20, we increase the pooling stride from 2 to 3, requiring the model to observe a larger matrix to deduce the pattern, thereby raising the demands on both visual reasoning and OCR proficiency. In each case, we adjust the test cases to align with the modified patterns introduced through the mutations, ensuring that the new tasks remain consistent with the requirements of our benchmark.

## C.4 ADDITIONAL DATASET STATISTICS

|  | dict | float | int | 1D list | 2D list | np.ndarray | str | tuple | pd.DataFrame | bool |
|---|---|---|---|---|---|---|---|---|---|---|
| Input | 8 | 3 | 34 | 35 | 24 | 2 | 4 | 12 | - | - |
| Output | - | 3 | 5 | 34 | 6 | 6 | 3 | 3 | 3 | 45 |

Table 9: The distribution of Input/Output types for the coding tasks in `HumanEval-V`.

The input and output (I/O) types used in the coding tasks in `HumanEval-V` are designed to maintain a low level of complexity. A distribution of their frequencies is shown in Table 9. We focus on using simple and commonly used data structures, such as integers, lists, dictionaries, and tuples, which are frequently encountered in standard programming tasks. Most of the tasks utilize basic types like integers, 1D and 2D lists, or simple boolean outputs, ensuring that solving them does not require specialized fine-tuning on domain-specific data. These I/O types are prevalent in open-source code

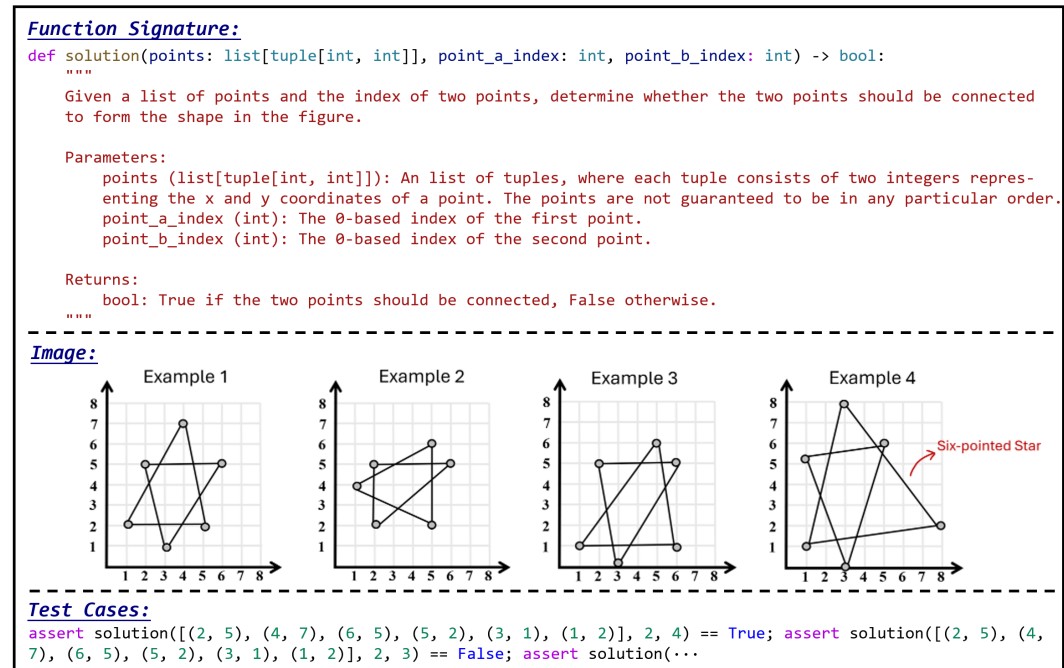

Figure 18: A mutated version of the coding task from Figure 15.

used for model pretraining, making our benchmark compatible with general-purpose LMMs without requiring additional adaptation or targeted training on specified datasets.

In terms of module dependencies, `HumanEval-V` utilizes a minimal set of common Python libraries, including `typing`, `pandas`, `numpy`, `math`, `heapq`, and `collections`. These libraries are well-supported and widely used in both general programming and scientific computing contexts. This ensures that our benchmark can comprehensively assess the visual reasoning capabilities of models using common and accessible libraries, without introducing dependencies that are rarely present in the training data. Notably, the coding tasks in `HumanEval-V` use only the stable APIs from these libraries, ensuring consistent and reliable testing.

## D  DETAILS OF THE EVALUATED MODELS

To facilitate the reproducibility of our results, we provide detailed information on all the evaluated models in Table 10. The open-weight models are sourced from Hugging Face[2], while the proprietary models are accessed via their respective APIs.

For model inference, we utilize 8 NVIDIA A800 GPUs and maintain the original tensor data types specified by each model to ensure consistent evaluation. To further optimize inference efficiency, we leverage the open-source framework vLLM[3].

Additionally, the Code LLMs used in Section 4.2 are also listed in Table 10. These models are fine-tuned versions of foundational LLMs, specifically trained on large-scale, multilingual programming datasets to enhance their performance across diverse coding scenarios.

## E  DISCUSSION ON THE MMCODE DATASET

MMCode (Li et al., 2024b) introduces a multimodal coding dataset aimed at evaluating LMMs' algorithmic problem-solving skills in visually rich contexts. The dataset includes 3,548 questions

---

[2]https://huggingface.co
[3]https://docs.vllm.ai/en/latest/

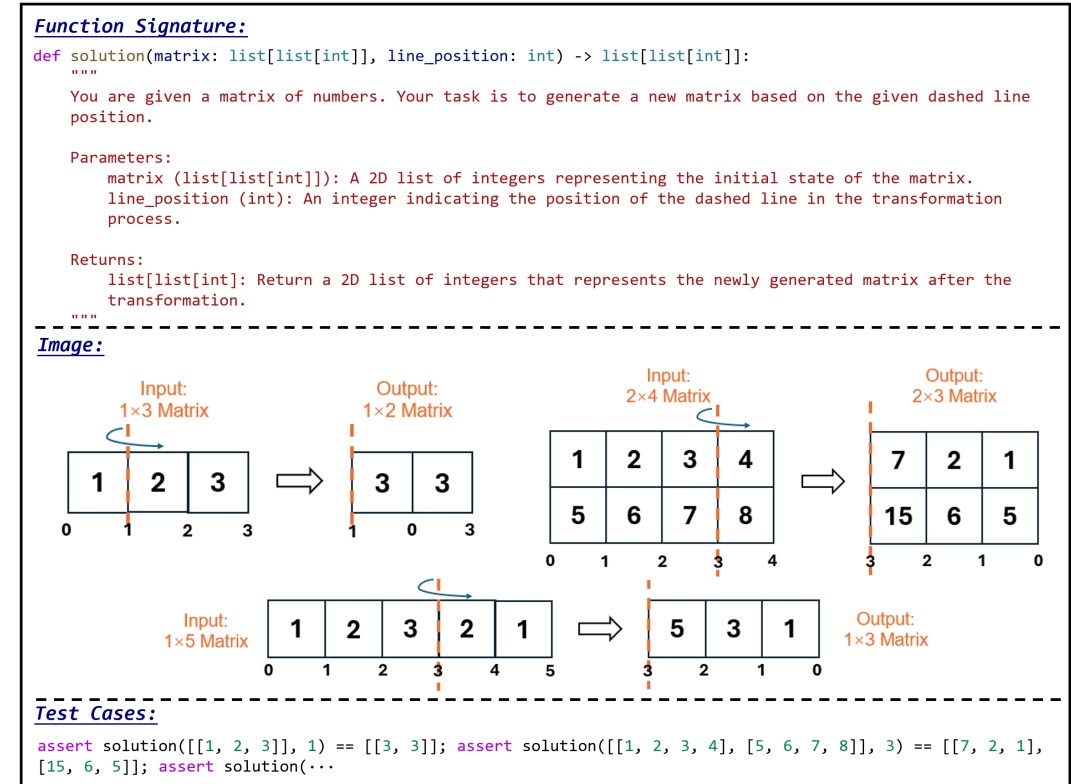

```
Function Signature:
def solution(matrix: list[list[int]], line_position: int) -> list[list[int]]:
    """
    You are given a matrix of numbers. Your task is to generate a new matrix based on the given dashed line
    position.

    Parameters:
        matrix (list[list[int]]): A 2D list of integers representing the initial state of the matrix.
        line_position (int): An integer indicating the position of the dashed line in the transformation
        process.

    Returns:
        list[list[int]]: Return a 2D list of integers that represents the newly generated matrix after the
        transformation.
    """
```

*Image:*

*Test Cases:*

```
assert solution([[1, 2, 3]], 1) == [[3, 3]]; assert solution([[1, 2, 3, 4], [5, 6, 7, 8]], 3) == [[7, 2, 1],
[15, 6, 5]]; assert solution(···
```

Figure 19: A mutated version of the coding task from Figure 16.

scraped from various competitive programming websites. However, as discussed in Appendix A, the issue of data leakage poses a significant challenge, as many of these coding tasks may have been previously encountered and memorized by the models, making them unsuitable for direct use as test data. Additionally, as demonstrated in Appendix C.1, a majority of the coding challenges in MMCode contain visual content that is redundant; the information conveyed through images can often be inferred from the textual descriptions alone, rendering the visuals non-essential. The reported results from MMCode further confirm this issue, as the performance using "language-only" inputs is similar to that with "vision + language" inputs.

In contrast, `HumanEval-V` is specifically designed to focus on visual understanding and reasoning abilities, rather than general coding proficiency, ensuring an irreplaceable dependency on visual context. During the annotation phase, we verify that language-only inputs achieve a 0% pass rate for GPT-4o, demonstrating the necessity of visual context in `HumanEval-V`. Moreover, our careful adaptation and mutation processes prevent data leakage, ensuring that evaluations accurately measure visual reasoning and coding abilities, rather than memorization of previously seen tasks.

## F  LIMITATIONS

Despite the contributions of our benchmark, several limitations remain that we aim to address in future work:

(1) Limited Number of Coding Tasks: The size of our benchmark is currently restricted due to the difficulty of identifying suitable coding problems and the challenges associated with adapting these problems to meet our standards. Each annotator has dedicated over 200 hours to constructing the benchmark, ensuring that every task is meticulously curated and aligns with our goals of testing visual reasoning. Our priority has been to maintain high quality, which we believe is crucial for deriving meaningful insights. Fortunately, the current version of `HumanEval-V` has already enabled us to identify several unique findings about the limitations of current LMMs. Moving forward,

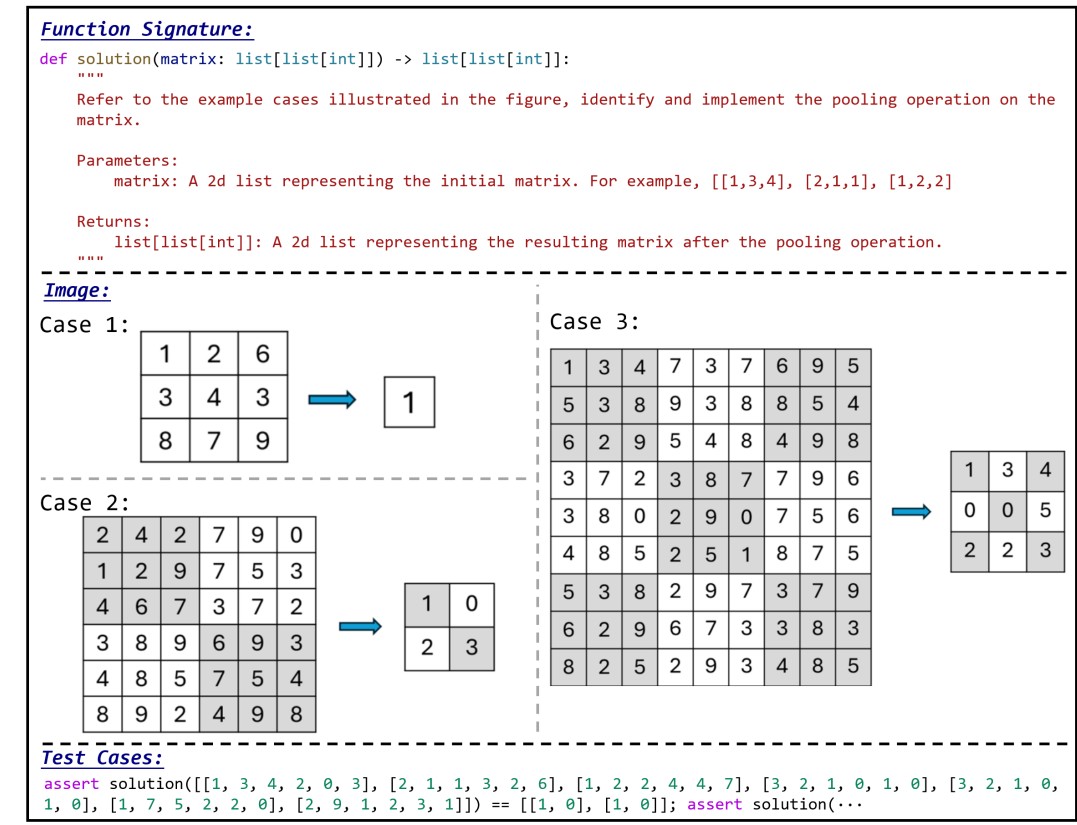

```
Function Signature:
def solution(matrix: list[list[int]]) -> list[list[int]]:
    """
    Refer to the example cases illustrated in the figure, identify and implement the pooling operation on the
    matrix.

    Parameters:
        matrix: A 2d list representing the initial matrix. For example, [[1,3,4], [2,1,1], [1,2,2]

    Returns:
        list[list[int]]: A 2d list representing the resulting matrix after the pooling operation.
    """
```

Image:

Case 1:

Case 2:

Case 3:

Test Cases:
```
assert solution([[1, 3, 4, 2, 0, 3], [2, 1, 1, 3, 2, 6], [1, 2, 2, 4, 4, 7], [3, 2, 1, 0, 1, 0], [3, 2, 1, 0,
1, 0], [1, 7, 5, 2, 2, 0], [2, 9, 1, 2, 3, 1]]) == [[1, 0], [1, 0]]; assert solution(···
```

Figure 20: A mutated version of the coding task from Figure 17.

we plan to expand `HumanEval-V` by continuing to annotate additional tasks using our established pipeline and guidelines. To benefit the community, we will open-source our annotation process and release all details of our work.

(2) Limited Model Coverage: While our experiments evaluate a diverse set of representative top-performing LMMs, the rapid pace of development in this area means that new models are frequently released, which may not be covered in our evaluation. We acknowledge that broader model coverage could provide a more comprehensive understanding of current capabilities. To address this, we will publicly release the evaluation toolkit and dataset, along with an up-to-date leaderboard to track ongoing advancements and benchmark new models as they become available. This will help keep our benchmark relevant and provide a platform for continuous assessment.

(3) Limited Scope of Experimental Analysis: Due to budget constraints, our in-depth analysis is limited to a subset of the evaluated models and hyper-parameter settings. While we have included as many models as possible to ensure that our findings are not biased toward specific LMMs, there are areas that remain unexplored, such as evaluating the impact of different prompting templates and experimenting with alternative sampling strategies, including varying temperature settings. Nevertheless, we have carefully chosen hyper-parameters that are widely used and deemed fair for cross-model comparisons. We believe that the settings used in our experiments provide reliable insights and lead to trustworthy conclusions. Additionally, our investigation into advanced reasoning methods is limited. In preliminary experiments, we applied the zero-shot Chain-of-Thoughts (CoT) (Wei et al., 2022) approach, which prompts the model to perform step-by-step reasoning before generating code. However, this method showed limited improvement in our coding tasks. Given that zero-shot CoT is a relatively weak baseline for reasoning research, fully exploring more sophisticated reasoning-enhancement techniques (Yao et al., 2024a; Mitra et al., 2024) would require significant resources. We leave this comprehensive study to future work.

| Models | Params | Links |
|---|---|---|
| | | **Proprietary** |
| GPT-4o-0513 | | https://platform.openai.com/docs/models/gpt-4o |
| GPT-4o-mini-0718 | | https://platform.openai.com/docs/models/gpt-4o-mini |
| Claude 3.5 Sonnet | | https://docs.anthropic.com/en/docs/about-claude/models |
| Gemini 1.5 Pro (001) | | https://ai.google.dev/gemini-api/docs/models/gemini |
| Gemini 1.5 Flash (001) | | https://ai.google.dev/gemini-api/docs/models/gemini |
| | | **Open-Weight LMM** |
| Qwen2-VL | 73.4B | https://huggingface.co/Qwen/Qwen2-VL-72B-Instruct |
| Qwen2-VL | 8.3B | https://huggingface.co/Qwen/Qwen2-VL-7B-Instruct |
| MiniCPM-V 2.6 | 8.1B | https://huggingface.co/openbmb/MiniCPM-V-2_6 |
| MiniCPM-V 2.5 | 8.5B | https://huggingface.co/openbmb/MiniCPM-Llama3-V-2_5 |
| InternVL-Chat-V1.5 | 25.5B | https://huggingface.co/OpenGVLab/InternVL-Chat-V1-5 |
| InternVL2 | 76.3B | https://huggingface.co/OpenGVLab/InternVL2-Llama3-76B |
| InternVL2 | 40.1B | https://huggingface.co/OpenGVLab/InternVL2-40B |
| InternVL2 | 25.5B | https://huggingface.co/OpenGVLab/InternVL2-26B |
| InternVL2 | 8.1B | https://huggingface.co/OpenGVLab/InternVL2-8B |
| InternVL2 | 4.2B | https://huggingface.co/OpenGVLab/InternVL2-4B |
| LLaVA-OneVision | 73.2B | https://huggingface.co/lmms-lab/llava-onevision-qwen2-72b-ov |
| LLaVA-OneVision | 8.0B | https://huggingface.co/lmms-lab/llava-onevision-qwen2-7b-ov |
| Phi-3.5-Vision | 4.2B | https://huggingface.co/microsoft/Phi-3.5-vision-instruct |
| Phi-3-Vision | 4.2B | https://huggingface.co/microsoft/Phi-3-vision-128k-instruct |
| | | **Open-Weight LLM** |
| Nous-Hermes-2-Yi | 34.4B | https://huggingface.co/NousResearch/Nous-Hermes-2-Yi-34B |
| InternLM2-Chat | 19.9B | https://huggingface.co/internlm/internlm2-chat-20b |
| InternLM2.5-Chat | 7.7B | https://huggingface.co/internlm/internlm2_5-7b-chat |
| Phi-3-Mini-Instruct | 3.8B | https://huggingface.co/microsoft/Phi-3-mini-128k-instruct |
| Phi-3.5-Mini-Instruct | 3.8B | https://huggingface.co/microsoft/Phi-3.5-mini-instruct |
| Qwen2 | 7.6B | https://huggingface.co/Qwen/Qwen2-7B |
| Llama-3-Instruct | 8.0B | https://huggingface.co/meta-llama/Meta-Llama-3-8B-Instruct |
| | | **Open-Weight Code LLM** |
| CodeStral | 22.2B | https://huggingface.co/mistralai/Codestral-22B-v0.1 |
| DSCoder-V2-Lite | 15.7B | https://huggingface.co/deepseek-ai/DeepSeek-Coder-V2-Lite-Instruct |
| Yi-Coder-Chat | 8.8B | https://huggingface.co/01-ai/Yi-Coder-9B-Chat |
| DSCoder-V1.5 | 6.9B | https://huggingface.co/deepseek-ai/deepseek-coder-7b-instruct-v1.5 |

Table 10: The model identification links.

