# OpenReview forum: "HumanEval-V: Evaluating Visual Understanding and Reasoning Abilities of Large Multimodal Models Through Coding Tasks"
_ICLR.cc/2025/Conference — ICLR 2025 Conference Withdrawn Submission_

### Official Review · Reviewer_W9sK · 2024-10-20

**Soundness:** 2
**Presentation:** 2
**Contribution:** 1
**Rating:** 3
**Confidence:** 4

**Summary:**

The paper presents HumanEval-V, a benchmark for evaluating Large Multimodal Models (LMMs) on entry-level coding tasks that require visual understanding and reasoning. It includes 108 adapted hand-written Python coding challenges, each with test cases. Current LMMs perform poorly in this benchmark.

**Strengths:**

- All questions are adapted from the original sources manually to avoid possible data contamination.
- Visual elements are critical in the questions.
- Extensive experiments on many code models.

**Weaknesses:**

- **Lack of novelty**: The benchmark and evaluation pipeline closely align with MMCode[1], without significant innovation.
- **Limited diversity of the benchmark**: the 108 questions are adapted from 40 core questions, and many images are reused across images (see questions)


[1] Li, K., Tian, Y., Hu, Q., Luo, Z., & Ma, J. (2024). MMCode: Evaluating multi-modal code large language models with visually rich programming problems. arXiv preprint arXiv:2404.09486.

**Questions:**

- Can you share more details on how the dataset was created? In Section 2.2,

> We then create a new coding problem by modifying the context and patterns of the original problem and redrawing the corresponding images.

To what extent did the authors modify the context and images? Do the adapted questions share a similar solution to the old questions?


> Following the initial annotation of the 40 coding tasks, we conduct an additional round of mutation-based extensions. This process expands the number of coding tasks based on the initial annotations, by creating similar yet distinct coding tasks.

> （in Appendix C.3) The objective is to generate new tasks that retain the essence of the original tasks but introduce
distinct patterns with minimal modification

Does this process hurt the diversity of the proposed dataset? How many distinct images are there in the dataset?

---

> ### Author Response · Authors · 2024-11-15
>
> ### W1: Lack of Novelty
>
> We respectfully disagree with the assertion that HumanEval-V lacks novelty. As discussed in our response to Reviewer ZHrF’s **W2**, HumanEval-V and MMCode have distinct purposes. MMCode primarily assesses **programming abilities** using coding problems scraped from competitive programming websites, where detailed textual descriptions are provided, and visual reasoning plays a minimal role. MMCode’s primary aim is to evaluate programming performance with additional visual inputs.
>
> In contrast, HumanEval-V specifically focuses on evaluating **visual understanding and reasoning abilities** in LMMs, using coding tasks as a structured means to assess these higher-order reasoning skills rather than programming abilities alone. This focus fills an important gap in the evaluation landscape, offering insights into LMMs' visual reasoning capabilities that are not the primary concern of MMCode.
>
> ---
>
> ### W2: Limited Diversity of the Benchmark
>
> As mentioned in our response to Reviewer aSN5’s **W1**, HumanEval-V was meticulously expanded through adaptation and mutation to enhance diversity. All images in HumanEval-V are redrawn and distinct, ensuring that adapted and mutated tasks independently challenge LMMs.
>
> ---
>
> ### Q1: Details on Dataset Creation and Adaptation Process
>
> Our annotation process involved **contextual modification** (e.g., altering object relationships, introducing new constraints) and **image redrawing** (e.g., changing object positions, adding visual elements) to create distinct tasks. The adapted questions are fundamentally different from the originals, and all images are redrawn to ensure uniqueness. For additional details, please refer to Appendix **C.2** and **C.3**.
>
> ---
>
> ### Q2: Dataset Diversity and Impact of Mutation-Based Extensions
>
> All 108 tasks feature unique images. To validate dataset diversity, we conducted analyses (discussed in the response to Reviewer aSN5 **W1**) showing that **success on one version of a task does not imply success on its mutated version**. Specifically, our pass@10 evaluations indicate that models solving an initial task have only a 40% probability of solving its mutated version within the same number of attempts. This low transfer rate demonstrates that adapted tasks are sufficiently distinct to challenge models independently.

---

### Official Review · Reviewer_ZHrF · 2024-10-27

**Soundness:** 3
**Presentation:** 3
**Contribution:** 2
**Rating:** 5
**Confidence:** 4

**Summary:**

The paper presents HumanEval-V, a benchmark designed to assess the visual reasoning capabilities of Large Multi-modal Models (LMMs) through code generation based on images. It comprises 108 entry-level Python coding tasks adapted from Codeforces and Stack Overflow. These tasks evaluate LMMs' ability to reason across both visual and textual contexts. The study examines 19 state-of-the-art LMMs, highlighting significant limitations in their visual reasoning and coding abilities, as well as the gap between advanced proprietary models and open-source models.

**Strengths:**

**Motivation & task design**: The motivation is clear, that it aims to eliminate any potential data leakage and strengthen the necessity of using image information for solving coding task. The paper demonstrates high-quality task design, ensuring that the visual context is essential for solving the coding problems.

**Clarity**: The dataset construction and evaluation are clearly articulated. The experimental settings and selected models are considered reasonable and comprehensive.

**Analysing experiments**: The analysing experiments are solid, demonstrating how models perform under different task settings.

**Weaknesses:**

**Limited Dataset Size**: Although the authors have acknowledged in the Limitations section that they plan to expand the dataset, the current version only includes 108 coding tasks, which were derived from a set of 40 tasks. This limited size raises concerns about the dataset's ability to comprehensively and robustly evaluate the full spectrum of visual reasoning & coding abilities in LMMs.


**Novelty Concerns**:  While the authors emphasize improvements over the MMCode benchmark, the degree of novelty in HumanEval-V remains limited. Both benchmarks focus on integrating visual elements with coding tasks. The contribution feels incremental rather than groundbreaking. There are also other coding related benchmark that emphasizes visual information like ChartMimic, Plot2Code.


**Applicability of the Task**: According to 2.1 or Appendix C.1, the authors sourced data from posts made from 2020, ultimately narrowing the dataset down to just 8 posts that met the criteria for inclusion in the benchmark. This raises concerns about the broader relevance and applicability of these tasks. If multi-modal coding tasks of this type are so rarely encountered, it calls into question whether this benchmark truly reflects the challenges that LMMs would face in real-world scenarios. The narrow focus may result in tasks that are too niche to provide meaningful insights into practical LMMs development.

**Questions:**

Apart from my main concerns listed in the above weaknesses part, here are some questions:
* Can the authors offer some analysis regarding the types of images on which LMMs perform better or worse? For example, are there specific visual patterns or image complexities (e.g., graphs, maps) where models consistently struggle or excel?
* The paper mentions that tasks were adapted and modified to prevent data leakage. Could the authors elaborate on the specific steps taken to ensure that LMMs do not rely on memorized patterns from previous coding datasets? Were any ablation studies conducted to verify the effectiveness of these modifications?
* The authors mention that human-annotated image descriptions significantly improved model performance. Could more details be provided on how these descriptions were structured?

---

> ### Author Response · Authors · 2024-11-15
>
> Thank you for your comments. We respond to each point below.
>
> ### W1: Limited Dataset Size
>
> As discussed in our response to Reviewer aSN5’s **W1**, HumanEval-V was designed to provide a lightweight yet high-quality benchmark, focusing on the specific task of evaluating **visual reasoning in LMMs**. While expanding dataset size is always beneficial, we prioritized rigorous curation and quality assurance to ensure each coding task met high standards for relevance, clarity, and difficulty. This trade-off allows HumanEval-V to efficiently and effectively expose visual reasoning limitations in current LMMs, even with a relatively small dataset.
>
> To create diversity, **HumanEval-V’s** 108 coding tasks are drawn from 40 unique cases that have been carefully adapted and mutated. Each task is redrawn to offer distinct perspectives on visual context comprehension. Our analyses show that model success on a single task does not guarantee success on its mutated variants, underscoring the effectiveness of this approach for robust evaluation.
>
> ### W2: Novelty Concerns
>
> We respectfully disagree with the view that HumanEval-V lacks sufficient novelty. While both HumanEval-V and MMCode incorporate visual elements with coding tasks, they have fundamentally different focuses. MMCode is primarily centered on **programming abilities**, often relying on existing problems scraped from coding competition websites, where textual descriptions are explicit, and visuals play a minor or even redundant role. MMCode thus emphasizes **programming performance** rather than visual reasoning.
>
> In contrast, HumanEval-V explicitly emphasizes **visual understanding and reasoning** as its central focus. Coding tasks serve as a structured and quantifiable way to assess these abilities. The benchmark challenges models to interpret visual contexts critically and requires models to extract essential information from images, making it significantly different from MMCode.
>
> Similarly, benchmarks like ChartMimic and Plot2Code focus on de-rendering visual elements into code, which mainly tests **perceptual abilities** such as OCR, object recognition, and parsing visual layouts. HumanEval-V differs in that its visual contexts involve complex, programmatic patterns, targeting higher-order **reasoning abilities** beyond simple visual recognition.
>
> ### W3: Applicability of the Task
>
> We believe that HumanEval-V provides insights into essential and broadly applicable visual reasoning abilities, which are relevant across many downstream tasks. The design choice to use coding tasks serves to maintain simplicity while making it easier for the research community to use the benchmark for repeatable, rigorous evaluations.
>
> Regarding the limited pool of cases from Stack Overflow, our focus was not on representing the diversity of real-world coding scenarios but on creating visual tasks that would require reasoning skills, not just perceptual skills. **Stack Overflow** posts include high-quality coding problems, but many involve complex context that would shift focus away from visual reasoning and towards more advanced programming tasks. To retain focus on visual understanding, we selected visually-driven coding tasks that could be solved with minimal textual descriptions and limited programming complexity.
>
> This does not mean that HumanEval-V lacks broader applicability. The visual reasoning tasks reflect essential abilities that LMMs must develop for diverse applications, from technical domains to general-purpose multimodal use cases. The relative rarity of suitable posts on Stack Overflow speaks to the fact that these reasoning challenges are too fundamental and intuitive for human users to ask, rather than niche for model evaluation.

---

> ### Author Response · Authors · 2024-11-15
>
> ### Q1: Analysis of Model Performance on Different Image Types
>
> The visual contexts in HumanEval-V go beyond simple types like graphs or maps, often involving compound objects and layered programmatic patterns, which add a level of abstraction. Consequently, performance analysis cannot be based solely on basic image types.
>
> Based on our analysis of experimental results, we found that current LMMs exhibit two main limitations in HumanEval-V tasks:
>
> 1. **Visual Complexity**: LMMs tend to perform better on tasks with simpler visual patterns and fewer elements. For instance, as shown in **Figure 4**, InternVL-2 (26B) struggled with an image illustrating the "OR" operation on 7-segment displays, where visual complexity was high, with many overlapping elements.
> 2. **Counterintuitive Contexts**: LMMs often struggle with tasks that include counterintuitive or unusual configurations. In **Figure 1**, for example, an image showing numbers arranged in a non-clockwise order created confusion for some models, indicating a gap in counterintuitive reasoning.
>
> ---
>
> ### Q2: Steps Taken to Prevent Data Leakage
>
> To prevent data leakage, as noted in **Line 192** of the paper, we ensured that GPT-4o could not solve any of the coding tasks without access to the images. This process confirmed that the visual context was critical for task completion, as well as ensuring that models could not rely on memorized patterns or prior knowledge alone. By focusing on image-driven tasks without detailed problem descriptions, we also prevented succeeding based purely on memorization or common coding patterns in existing datasets.
>
> ---
>
> ### Q3: Structure of Human-Annotated Image Descriptions
>
> Please refer to **Appendix B.3** for a detailed illustration of the image description annotation process. In this section, we outline the structure of the annotations and provide examples to clarify how descriptions were created to convey relevant visual information without revealing solutions directly.

---

### Official Review · Reviewer_FGJT · 2024-10-29

**Soundness:** 2
**Presentation:** 3
**Contribution:** 2
**Rating:** 5
**Confidence:** 4

**Summary:**

The paper introduces HumanEval-V, a novel benchmark designed to evaluate the visual understanding and reasoning capabilities of Large Multimodal Models (LMMs) through coding tasks. It addresses a gap in existing benchmarks by focusing on tasks that require both visual reasoning and coding abilities. The benchmark comprises 108 entry-level Python coding tasks that necessitate visual context to solve, adapted from platforms like CodeForces and Stack Overflow to prevent data leakage. Each task is equipped with handcrafted test cases for thorough evaluation. The paper reports the results of 19 state-of-the-art LMMs on HumanEval-V, revealing significant challenges in current LMMs' visual reasoning and coding abilities, with even leading models achieving low pass rates. Ablation studies demonstrate performance gains when models are provided with textual descriptions of images, indicating the need for enhanced visual understanding capabilities. The findings highlight areas for future research to improve LMMs' visual reasoning and coding skills.

**Strengths:**

- Novel Benchmark: The paper introduces HumanEval-V, a unique benchmark that specifically targets the visual understanding and reasoning capabilities of LMMs through coding tasks, addressing a significant gap in current evaluation methods.
- Comprehensive Evaluation: Each task is equipped with handcrafted test cases, allowing for a thorough and reliable evaluation of the model-generated code solutions.

**Weaknesses:**

- Limited Number of Coding Tasks: The benchmark currently contains a relatively small number of tasks, which may limit the breadth of the evaluation. But the construction of this dataset does require a lot of human effort.
- This work evaluates the reasoning ability of MLLM from a new perspective, but I think this work is slightly simpler and less workload.

**Questions:**

No questions.

---

> ### Author Response · Authors · 2024-11-15
>
> Thank you for your review. We address each of your comments in detail below.
>
> ### W1: Limited Number of Coding Tasks
>
> As we discussed in our response to **Reviewer aSN5’s W1**, HumanEval-V is purposefully designed to be **a lightweight yet high-quality** benchmark focused on assessing the visual reasoning abilities of LMMs. While expanding the dataset size could add breadth, we prioritized ensuring each task met rigorous standards for quality, consistency, and visual reasoning complexity. We dedicated extensive human effort to curating tasks that would be challenging, distinctive, and free of data leakage while retaining a manageable scope for efficient testing. Our experiments, supported by statistics, showing that these tasks introduce enough variation to provide valuable insights into the strengths and weaknesses of current LMMs.
>
> ### W2: Benchmark Complexity and Effort
>
> We respectfully disagree with the notion that HumanEval-V is too simplistic or lacking in workload. Our benchmark is focused on **visual understanding and reasoning** within LMMs, and we have invested substantial effort to ensure that HumanEval-V meets this focus in a rigorous, valuable manner for the research community.
>
> Achieving this high standard involved overcoming many challenges:
>
> 1. **Visual Context Relevance**: Each task’s visual component is critical to solving the problem. Images are designed to contain all relevant information needed for task completion, ensuring that visual context is not extraneous but essential.
> 2. **Self-Explanatory Tasks**: Each coding task was developed to be primarily self-explanatory through its visual context, with minimal reliance on textual descriptions.
> 3. **Entry-Level Coding Complexity**: The coding tasks are aimed at an entry-level programming level, requiring only common Python libraries, to keep the focus on reasoning rather than programming difficulty.
>
> As detailed in **Appendix C**, we share the specific challenges and solutions we encountered during the benchmark’s construction. This effort not only ensures HumanEval-V’s quality but also provides a useful reference for the research community on conducting rigorous multimodal evaluations that reveal critical weaknesses in LMMs.
>
> Furthermore, we have conducted extensive **evaluations and ablation studies** across a diverse set of proprietary and open-weight models, analyzing correlations with other benchmarks. These analyses yielded findings that we believe will offer valuable insights and directions for future research in multimodal model development.

---

### Official Review · Reviewer_aSN5 · 2024-11-01

**Soundness:** 3
**Presentation:** 4
**Contribution:** 2
**Rating:** 5
**Confidence:** 3

**Summary:**

The paper introduces a multi-modal code generation benchmark, HumanEval-V, containing 108 test samples. Unlike traditional code generation tasks, this benchmark allows images to serve as the problem input (e.g., charts, trees, maps). The authors benchmarked a variety of both closed and open large multimodal models, finding that while baseline performance is generally low (e.g., GPT4o at 13% pass@1), providing annotated image descriptions significantly boosts performance (e.g., GPT4o from 13% to 45.4%).

**Strengths:**

- This benchmark effectively assesses both visual reasoning and code generation capabilities, providing a cohesive indicator for multi-modal models, especially those adapted from LLMs with vision encoders. Balancing textual and visual reasoning is challenging in training, and this benchmark helps to evaluate these combined strengths.
- The benchmark demonstrates a high level of curation and screening, reducing data contamination and ensuring that the input image is essential to solving the task.
- The writing is well-organized and detailed, enhancing readability.

**Weaknesses:**

- The benchmark’s size (108 test samples) and real-world coverage are somewhat limited, as these samples originate from 40 unique cases. While curated, the small sample size might limit generalizability and increase susceptibility to overfitting. Expanding the dataset with more diverse sources and formats beyond traditional coding puzzles could improve robustness. For example, a task might involve writing the code behind the given plot and changing its top line color from blue to green.
- While the benchmark is valuable, combining multimodal reasoning with code generation might seem niche. While beneficial to evaluate both capabilities together, the community can also test them independently. It remains to be clarified whether this benchmark is essential for multimodal or code generation research areas.

**Questions:**

On line#472, Qwen-VL LMM seems to outperform LLM by a large margin (6.7%) but not specified in “while InternVL-2 (4.2B) and LLaVA-OneVision (8B) show the least”. Adding a bit more context could help prevent potential reader misinterpretation.

---

> ### Author Response · Authors · 2024-11-15
>
> Thank you for your comments. Below, we address each of your concerns in detail.
>
> ### W1: Benchmark Size and Real-World Coverage
>
> HumanEval-V was designed to be lightweight and efficient, serving as an informative measure of **LMMs’ visual reasoning abilities**. We prioritized **quality over quantity**, ensuring each coding task met specific standards: tasks were curated and redrawn with no data leakage, at an entry-level coding difficulty, and with clear dependence on visual context. HumanEval-V includes a broad variety of visual elements like trees, graphs, maps, plots, and beyond. Each visual context in the benchmark was created to be self-explanatory and rich in information, embedding complex rules, conditions, and algorithmic patterns into single images. This high standard requires significant design effort, but offers great value for the evaluation of LMMs.
>
> Moreover, HumanEval-V has demonstrated its effectiveness in benchmarking multimodal models by exposing key performance gaps between open-weight and proprietary models and yielding valuable insights into the limitations of current LMMs. The **benchmark’s diversity** in both visual and coding patterns has proven valuable in uncovering weaknesses in LMMs, validating its design as a focused, high-quality tool for multimodal evaluation.
>
> We also recognize the importance of dataset size. HumanEval-V’s 108 coding tasks are derived from 40 carefully screened unique cases, with each case mutated and redrawn to create diversity and robustness. Significant effort was put into ensuring that each mutated task presented a distinct perspective on visual context comprehension. To assess the effectiveness of our mutation process, we analyzed pass@10 results across models like **GPT-4o, GPT-4o-mini, Claude 3.5 Sonnet,** and **Gemini Pro**. Results showed that if a model could solve a coding task within 10 attempts, the probability of solving its mutated version within 10 attempts was only 40%. This outcome confirms the effectiveness and diversity of the mutation process, expanding the benchmark’s ability to reveal gaps in multimodal model performance.
>
> Regarding your suggestion to include tasks with variations in task types, such as color modification in charts, we would like to clarify that HumanEval-V is not a traditional coding benchmark. Its core objective is to evaluate **visual reasoning and understanding**, with coding used as a proxy to assess complex reasoning. The suggested task (changing a line color) primarily tests perception and familiarity with plotting tools, which falls outside the focus of our study.
>
> ### W2: Combining Multimodal Reasoning and Code Generation
>
> We respectfully disagree that HumanEval-V’s integration of image understanding and coding represents a niche focus. As discussed in our response to Reviewer **oUCU’s** comment **W1**, the benchmark emphasizes **visual understanding and reasoning**, using coding as a valuable medium to assess these abilities in a structured and quantifiable way. This combination allows HumanEval-V to address previously uncovered aspects of LMM evaluation, advancing the field in assessing the alignment between visual and reasoning competencies in a rigorous coding task setting. We believe this benchmark provides a meaningful and unique contribution to the multimodal research community.
>
> ### Q1: Clarification on Qwen-VL’s Performance on Line #472
>
> As observed in **Table 4**, Qwen2-VL exhibits some inconsistencies in its coding abilities compared to its LLM counterpart, Qwen2. On HumanEval+, Qwen2-VL achieves a 6.7% improvement, whereas, on MBPP+, its performance **drops by 9.5%**. These contrasting results highlight variability in multimodal model performance, which aligns with our findings on the challenges these models face across different tasks. We will incorporate additional context around Qwen-VL’s performance to ensure clarity and prevent any potential misinterpretation by readers.

---

### Official Review · Reviewer_oUCU · 2024-11-01

**Soundness:** 3
**Presentation:** 3
**Contribution:** 2
**Rating:** 5
**Confidence:** 3

**Summary:**

The paper HumanEval-V primarily assesses the performance of large multimodal models (LMMs) in tasks incorporating visual understanding and code generation. The models show significant challenges in generalizing across multimodal inputs and reasoning through intricate tasks. Furthermore, the models' preference for simpler tasks and the difficulty in objectively evaluating visual reasoning capabilities limit their practical applications.

However, there are certain limitations. The benchmark's rigid code-image integration limits authentic assessment of multimodal code generation, and score differentiation is minimal. Additionally, the narrow evaluation scope lacks detailed metrics, restricting insights into the models' full capabilities.

**Strengths:**

Key strengths of this paper:
1. its introduction of a new benchmark, HumanEval-V, specifically designed to evaluate the visual understanding and reasoning capabilities of multimodal models—an area previously lacking in systematic evaluation standards
2. By using coding tasks to test models’ ability to process visual information. Additionally, it also assesses the alignment between multimodal and language understanding capabilities.

**Weaknesses:**

1. Rigid integration of code and image tasks: Many tasks do not necessarily require code-based solutions, making it stiff to reflect the code generation abilities of multimodal models or how multimodal information enhances code generation. This results in a somewhat stiff.
2. Limited differentiation in performance scores: Many models achieve near-zero scores, making it difficult to effectively distinguish performance levels between models.
3. Narrow evaluation scope: The evaluation lacks detailed metrics for multimodal understanding and code generation, overlooking the potential of multimodal models to assist in understanding code tasks or provide feedback, resulting in a singularity assessment of model capabilities.

**Questions:**

1. In the experiments, how do you understand the fact that image + explanation often yields poorer results than using explanation alone? Have you adjusted the prompts? Can images not enhance understanding?
2. Are there multi-image question types included?
3. Why is the code capability weak even when only given direct explanations? Do you know the reasons behind this? Its performance on HumanEval+ and MBPP+ appears normal.
4. Why not consider using images as feedback signals?

---

> ### Author Response · Authors · 2024-11-14
>
> We appreciate your detailed feedback. Below, we address each of your comments and provide clarifications to support our choices and findings.
>
> ---
>
> ### W1: Rigid integration of code and image tasks
> You mention that integrating code with image tasks seems rigid, and that it doesn't reflect the full potential of multimodal models in code generation. We would like to clarify that HumanEval-V is focused on evaluating the **visual understanding and reasoning abilities** of large multimodal models (LMMs), not their code generation capabilities. While many multimodal models excel in general knowledge and common-sense reasoning tasks, HumanEval-V presents coding tasks that demand higher-level reasoning and visual context interpretation, which are not typically addressed by current benchmarks. The integration of visual elements with coding tasks is intentional, as it evaluates the ability of models to **reason over visual patterns that requires programmatic thinking**. We believe this approach provides valuable insights that other benchmarks overlook.
>
> ---
>
> ### W2: Limited differentiation in performance scores
> While we acknowledge that many smaller-scale LMMs achieve near-zero scores, we strongly disagree with the characterization that HumanEval-V lacks performance differentiation. It is true that smaller models with limited capabilities often perform poorly, but this highlights the **unique challenge** that HumanEval-V offers in assessing visual reasoning abilities. For example, in **Table 7**, we show how Qwen2-VL (8.3B) achieves near-zero performance in HumanEval-V, whereas it performs well on other benchmarks like MMMU and MathVista. This stark contrast emphasizes the **gaps in visual and reasoning capabilities** that the current benchmarks cannot capture.
>
> Additionally, we have intentionally designed HumanEval-V to be **approachable**, with coding tasks that can be solved by entry-level programmers (as shown in **Table 1**). These tasks require only short context lengths and relatively simple code (median ground truth solution: 14 lines), so the low performance of smaller models provides valuable insights for improvement.
>
> Further, **Table 5** and **Table 6** demonstrate that performance improves significantly with more examples, which suggests ample room for model advancement. For instance, InternVL-2 (4.2B) reaches 14.8% pass@100, and GPT-4o reaches 68.5% pass@1000. These results underscore that HumanEval-V identifies critical weaknesses in current models, offering crucial directions for future work.
>
> Thus, we do not consider the near-zero scores of smaller models as a weakness; rather, they reveal important areas for improvement and contribute valuable insights for the research community.
>
> ---
>
> ### W3: Narrow evaluation scope
>
> We understand your suggestion of expanding the evaluation to consider multimodal understanding and agentic workflows. However, the tasks in HumanEval-V are designed to be **simple and lightweight**, with the goal of assessing zero-shot performance on entry-level coding problems. This approach mirrors other widely-used benchmarks like HumanEval and MBPP, which evaluate LLMs' coding abilities without the added complexity of agentic behavior.
>
> Incorporating agentic evaluation would introduce significant complexities, such as the need for sophisticated prompt engineering and multi-round inference, which would likely increase computation costs. Given the simplicity and zero-shot nature of the current tasks, we felt that focusing on visual reasoning and code generation abilities was more in line with the goals of the benchmark.
>
> That being said, we recognize the importance of investigating fine-grained abilities, and our **ablation studies** (Section 4.2) provide a more granular analysis of LMMs' abilities in visual understanding and code generation. These studies aim to highlight the model’s strengths and weaknesses in specific aspects.
>
> ---
>
> ### Q1: Image + explanation yielding poorer results
>
> We respectfully disagree with the conclusion that "image + explanation often yields poorer results than using explanation alone." In fact, **Table 3** shows that for **7 out of 12 models**, performance improved when both images and textual descriptions were provided. This indicates that for many models, combining visual and textual inputs does lead to performance gains.
>
> However, there are cases where the inclusion of images can hurt performance. This can occur when models fail to correctly interpret the visual elements, leading to **hallucinations** or incorrect assumptions about the image content. In these cases, the multimodal input may not provide any additional benefit, and might even hinder performance due to the model's incorrect interpretation of the visual context. Since the image descriptions are human-annotated, we can be confident that the textual information is accurate and grounded. And in such cases, relying solely on textual descriptions can be more reliable.

---

> > ### Author Response · Authors · 2024-11-14
> >
> > ### Q2: Multi-image question types
> >
> > As indicated in **Line 75** of the paper, all tasks in HumanEval-V involve only **one image**. This design choice keeps the benchmark simple and focused on evaluating visual understanding and reasoning capabilities, rather than multi-image processing.
> >
> > ---
> >
> > ### Q3: Weak code generation even with direct explanations
> >
> > The weak performance in code generation, even with direct textual explanations, can be attributed to two factors:
> >
> > 1. Visual Understanding: Textual descriptions, while helpful, are often insufficient for fully conveying the necessary context of the images. Models need to reason over the visual content, not just the textual explanation, to generate accurate code. The limitation here is primarily in the **model's ability to reason about visual information**.
> > 2. Degradation in Coding Ability: As shown in **Table 4**, most LMMs exhibit degraded performance in coding benchmarks like HumanEval+ and MBPP+, especially in the case of smaller models. This suggests that the models may struggle with more complex tasks that require both visual understanding and programmatic reasoning.
> >
> > ---
> >
> > ### Q4: Using images as feedback signals
> >
> > We did not incorporate images as feedback signals in this study, as we did not test agentic evaluation workflows. As discussed in **W3**, adding feedback loops would complicate the evaluation process significantly, adding complexity for prompt engineering and computation cost. We chose to keep the evaluation focused on zero-shot performance to maintain simplicity and clarity. And we explore the fine-grained abilities of LMMs through ablation studies.

---

### Note · Authors · 2024-11-15

I have read and agree with the venue's withdrawal policy on behalf of myself and my co-authors.